# Shifting winter atmospheric teleconnections to the North Pacific reconcile Younger-Dryas and Holocene $\delta^{18}O$ signals

Lesleigh Anderson[1], Bruce P. Finney [2,3] & W. Brad Baxter[3]

Using Alaskan lake sediment oxygen isotope records ($\delta^{18}O$), which trace the $\delta^{18}O$ of precipitation, we establish that abrupt atmospheric shifts occurred during the last deglacial period in the North Pacific-Arctic. The robust lake $\delta^{18}O$ chronologies confidently correlate Younger-Dryas (YD) atmospheric adjustments in Alaska with Greenland ice-core records and their seasonal sensitivity are consistent with cooling during winter. In contrast, abrupt $\delta^{18}O$ decreases during the late Holocene observed in our records, of similar magnitude as the YD, are best explained by atmospheric modes involving long-distance transport of sub-tropical Pacific moisture. Our sediment cores are among the most reliably dated records yet produced in the circum-Arctic and show that similar decreases in $\delta^{18}O$ of winter precipitation during the YD and late Holocene were driven by different atmospheric teleconnections. These results underscore major roles for seasonality and atmospheric patterns in the conceptual understanding of global scale climate oscillations, both past and future.

Understanding the geographic extent of reorganizations in atmospheric circulation recognized in the Greenland ice core and other North Atlantic records during stadial and interstadial periods is essential for improving our conceptual models of atmospheric teleconnections with far-reaching impacts on regional climate systems across the globe. However, key uncertainties persist regarding the nature and timing of deglacial climate changes in the North Pacific-Arctic sector, and how they are related to forcing from the North Atlantic. For example, during the Younger Dryas (YD), cooling in Greenland, driven by ice-sheet meltwater discharge and weakened North Atlantic Deep Water (NADW) formation, is expected to have triggered synchronous changes in the North Pacific-Arctic driven by an intensified jet stream and expanded polar front[1,2]. However, this connection has yet to be confidently demonstrated[3]. Here, we present precisely dated records on the oxygen isotopes of Alaska precipitation ($\delta^{18}O_p$) that are comparable with Greenland ice-core records, which enhances understanding of deglacial and Holocene atmospheric adjustments across the Northern Hemisphere.

We hypothesize that winter cooling during the YD would result in synchronous and abrupt decreases in $\delta^{18}O_p$ in both the North Atlantic and North Pacific regions due to temperature of condensation-related Rayleigh distillation. In contrast, changes in the $\delta^{18}O_p$ of Alaskan records that do not follow variations observed in the North Atlantic require a different climatic mechanism. Here we explore the idea that different winter atmospheric teleconnections to the North Pacific during two contrasting time intervals (e.g., deglacial and late Holocene) can result in similar decreases in $\delta^{18}O_p$ values in Alaska. We postulate that as ice sheets melted and Alaska's climate became increasingly maritime, its atmospheric connection to the North Atlantic weakened and became more strongly influenced by the subtropical North Pacific as the intensity and frequency of El Niño Southern Oscillation (ENSO) increased in response to orbital precession[4–7]. Testing our hypotheses of shifts in winter atmospheric teleconnections requires winter sensitive proxies, which are typically underrepresented in proxy records, but are the focus of this study.

Our work builds on the strengths of existing records[8–13] by improved record chronology, continuity, and temporal resolution provided by thick authigenic carbonate sediment packages within marl lakes in Upper Cook Inlet, that formed within glacial deposits following ice retreat within the Matanuska and Susitna Valley (MatSu; Fig. 1) by

[1]United States Geological Survey, Geosciences and Environmental Change Science Center, Denver, CO, USA. [2]Department of Biological Sciences, Idaho State University, Pocatello, ID, USA. [3]Department of Geosciences, Idaho State University, Pocatello, ID, USA. ✉e-mail: finney@isu.edu

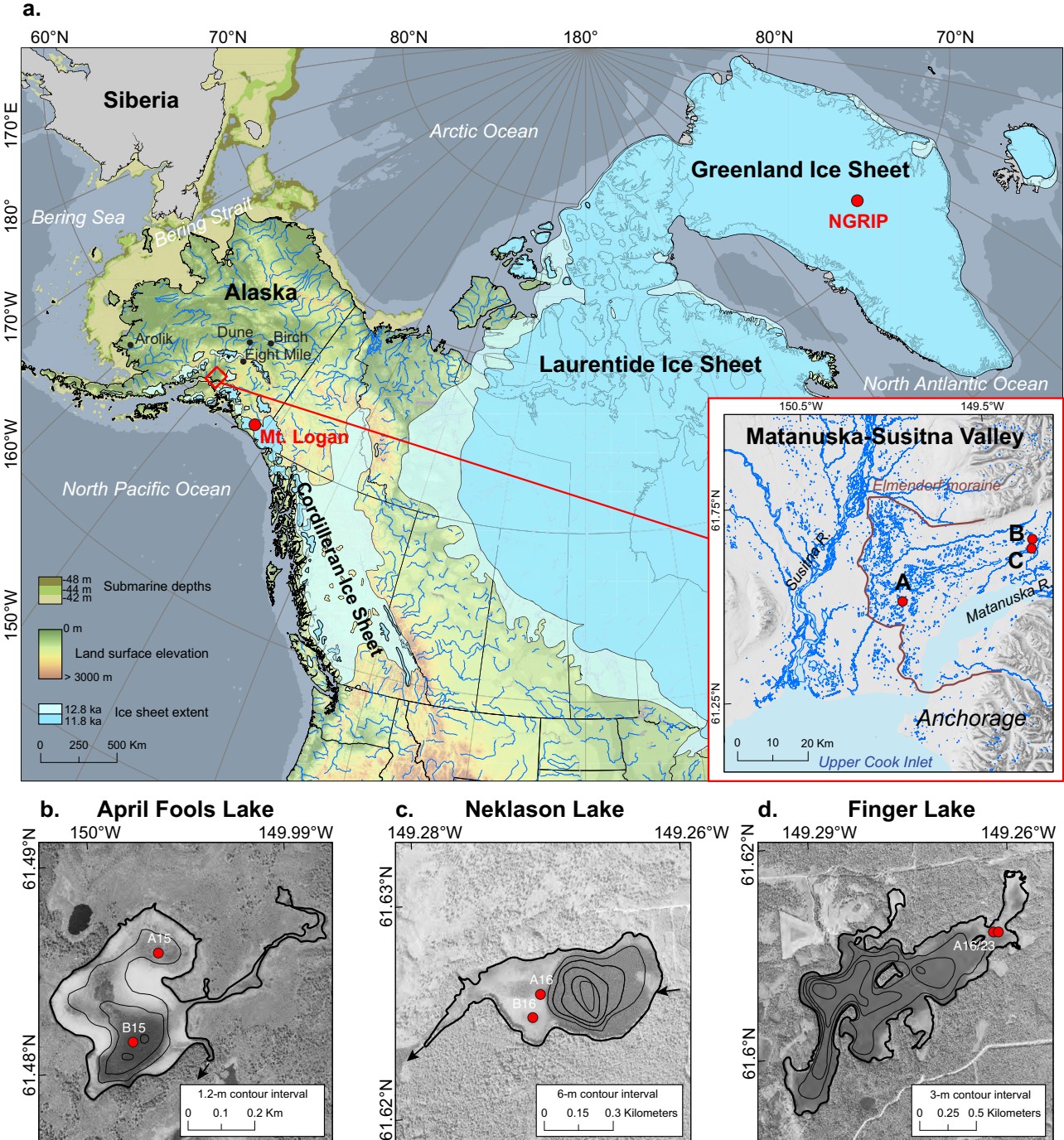

**Fig. 1 | Location map and Alaskan lake study sites. a** Laurentide, Cordilleran, and Greenland ice sheet margins are shown at 12.8 ka (light blue) and 11.8 ka (dark blue)[77,78] along with submarine depths of the present sills in the Bering Strait[47], and locations of lake sediment and ice core records of $\delta^{18}O$ (red), and existing lake sediment organic $\delta^{13}C$ records (black). Map inset of the Matanuska and Susitna Valley shows Elmendorf moraine extent at ~17ka[14] and our study lake locations. Study lake bathymetry and sediment core locations are shown on gray shade historic aerial photography (acquisition year 1950 in (**b**) and 1953 in (**c**, **d**), courtesy of the United States Geological Survey). The light gray areas in shallow lake margins indicate marl surface sediments that are visible through clear lake water.

~16.4 ka[14] (calibrated kiloyears before present). High rates of lake sediment accumulation mediated by biological productivity were dated by several methods to provide strong age-depth certainty, including radiogenic lead-210 ($^{210}Pb$), radiocarbon dating of terrestrial macrofossils, and tephrochronology (see Methods and Supplementary Fig. 5 and Supplementary Table 7). The closely sampled bulk carbonate oxygen isotopes ($\delta^{18}O_c$) provide high temporal resolution at multi-annual to centennial scales (see Methods and Supplementary Fig. 3 and Supplementary Table 6). The robust bulk sediment carbonate isotope proxy, consistently of authigenic origin (i.e., carbonate formed within lake), records the $\delta^{18}O$ of lake water (see Methods and Supplementary Fig. 2 and Supplementary Table 4).

Detailed hydrogeologic studies established that MatSu lakes are sourced by regional groundwater that is dominantly recharged by snowmelt originating as winter precipitation in the surrounding mountains of the ~1500 km² area[15] (Fig. 1). Accordingly, this baseline groundwater $\delta^{18}O$ signal is a proxy for winter $\delta^{18}O_p$ that is transmitted to all MatSu lakes, which in turn have different open and closed

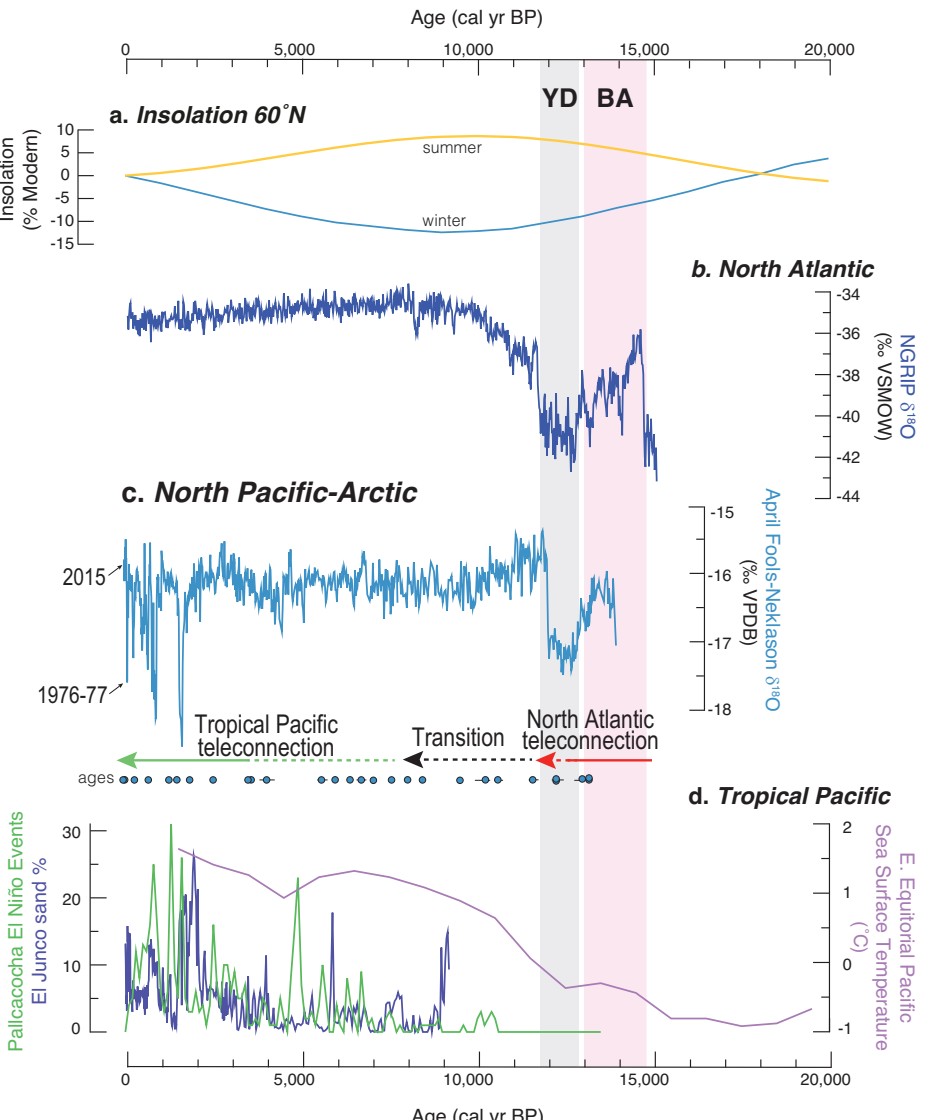

**Fig. 2 | Alaska lake bulk carbonate δ¹⁸O and Greenland ice-core δ¹⁸O. a** Solar seasonal insolation (60°N, % difference from modern) are shown with **b**, Greenland ice-core δ¹⁸O (NGRIP[79,80], dark blue), and **c** North Pacific-Arctic δ¹⁸O_p from Alaskan lake carbonate δ¹⁸O (this study, blue circles indicate age points, 1-sigma range shown if outside symbol size). **d** Tropical Pacific records of El Niño variability from lakes Pallcacocha[6] and El Junco[57] are shown with eastern equatorial Pacific Ocean composite alkenone sea surface temperature index[4]. YD Younger Dryas, BA Bølling Allerød. Horizontal arrows summarize proposed shifts in winter atmospheric teleconnection.

hydrologic controls that generate lake-water-δ¹⁸O variations more sensitive to winter or summer conditions[16] (see Methods and Supplementary Fig. 1 and Supplementary Table 1). We selected hydrologically open and closed lakes for our sediment δ¹⁸O_c reconstructions to provide unique perspectives on changes in seasonality through time. Our open study lakes, April Fools Lake and Neklason Lake, have records of lake-water δ¹⁸O sensitive to winter δ¹⁸O_p. Their combined sediment δ¹⁸O_c records overlap with the modern range of δ¹⁸O_p between −15 to −19‰[17] (Fig. 2c, see Methods and Supplementary Fig. 7). Our closed study lake, Finger Lake, has lake-water δ¹⁸O sensitive to summer evaporation resulting from long lake-water-residence times that lead to ¹⁸O-enrichment, overprinting the winter δ¹⁸O_p[16]. Thus, the higher δ¹⁸O_c values in Finger Lake, between −10 to −15‰, relative to our open lake records, reflect summer temperature and aridity conditions that vary independently from winter[17].

The regional source of precipitation and prevailing winds in south-central Alaska today is southerly (-75%) modulated by the Aleutian Low (AL) semi-permanent pressure system that intensifies

during winter. Surrounding mountain ranges (>4000 m elevation) control spatial precipitation patterns by orographic uplift[18] and coastal topographic barriers orient upper-level geostrophic flow; a process that was maintained by continental ice sheet barriers during past glacial periods[19,20]. Observations have shown that relatively low δ¹⁸O_p values are linked to stronger frontal boundary interactions, which enhance topographic rainout and mountain snowfall[21]. Strong frontal boundaries are most commonly driven by meridional flow patterns that transport warm moist air from lower latitudes in the Pacific Ocean resulting in long-distance rainout during vapor transport. Meridional patterns are associated with an intensified or eastward positioned AL that facilitates northward moisture transport during Pacific climate modes, such as the positive modes of the Pacific North America pattern (+PNA), the Pacific Decadal Oscillation (+PDO), and the El Niño Southern Oscillation (El Niño)[22,23]. In contrast, relatively high δ¹⁸O_p values are linked with zonal flow patterns and negative Pacific modes (-PNA, -PDO, and La Niña). Zonal patterns are characterized by a weakened or westward positioned AL wherein moisture is more frequently

derived from cooler northern ocean sources, including the Bering Sea. Less rainout across shorter vapor transport distances combined with cooler airmasses also results in less mountain snowfall[16].

Temporal changes in contemporary $\delta^{18}O_p$ in southern Alaska reflect strong relationships with winter atmospheric circulation patterns, weakly related to average temperature[16,17,24]. These observations are confirmed in our records by an abrupt $\delta^{18}O_c$ decrease at April Fools Lake during the positive PDO shift during the winter of 1976–77 which contributed to winter warming and increased precipitation in south-central Alaska (Fig. 2)[25,26]. At our closed lake, a strong summer evaporation signal resulted in a muted $\delta^{18}O_c$ response to the atmospheric circulation shift[16]. Previous proxy studies over the Holocene of North Pacific $\delta^{18}O_p$ support atmospheric circulation controls and show weak correlations with instrumental temperature records or paleo-temperature reconstructions[27–29]. For example, higher $\delta^{18}O_p$ in the Mt. Logan ice core during the Little Ice Age compared to the 20th century would suggest warmer conditions if driven by temperature, which is inconsistent with the cold period[13]. A more likely explanation for higher $\delta^{18}O_p$ values is a northward shift in moisture source and shorter vapor transport pathways[30].

In this study, we identify similar winter $\delta^{18}O_p$ anomalies in the North Pacific region during the Younger Dryas and the late Holocene in spite of different global boundary conditions. To reconcile these observations we attribute a shift in dominant winter atmospheric teleconnections from the North Atlantic to the sub-tropical Pacific. This reorganization likely reflects changes in large-scale circulation patterns, including the Aleutian Low and polar jet, which modulate moisture transport into the North Pacific. By providing continuous records from the last glaciation to present for the North Pacific-Arctic region, our Alaskan lake sediment $\delta^{18}O_c$ data allow for confident chronological assessment of seasonal climatic mechanisms with implications for understanding the sensitivity of the North Pacific to shifts in ocean-atmosphere dynamics and global linkages.

## Results and discussion
### North Atlantic controls on Alaska $\delta^{18}O_p$ during the Younger-Dryas

During regional deglaciation, our lake sediment records indicate early lake development within the Bølling-Allerød (BA) chronozone and intermediate $\delta^{18}O_c$ values with respect to their range over the full record (Fig. 2). Beginning ~12.8 ka, a sharp decrease in $\delta^{18}O_c$ is synchronous with onset of the YD chronozone documented in Greenland ice cores within the uncertainty of the lake age models (see Supplementary Fig. 5 and Supplementary Table 6). During the YD chronozone, Alaska $\delta^{18}O_c$ values were continuously at the lowest levels of the deglacial transition, followed by an abrupt increase of +2‰ at ~11.7 ka. This simultaneous $\delta^{18}O$ variation in sediment cores from Alaskan lakes and Greenland ice substantiates an atmospheric temperature teleconnection between Alaska with the North Atlantic driven by variations in Atlantic Meridional Overturning Circulation (AMOC).

A temperature-based interpretation of the decrease of $\delta^{18}O_p$ in Alaska during this period is supported by (1) global boundary conditions that shifted the polar jet stream southward[31]; (2) atmospheric circulation characterized by zonal flow over colder Gulf of Alaska sea surface temperatures[32,33], expanded Bering Sea ice cover[34] and a weakened AL[35]; and (3) reconstructions indicating dampened ENSO activity during deglaciation by contracted sub-tropical zonal sea surface temperature gradients[4]. We estimate air temperature declines in Alaska during the YD ranged from ~4 to 8 °C based on the combined effects of $\delta^{18}O$ temperature fractionation effects during water vapor condensation and by lake calcite formation[11] and considering the influence of sea surface temperature variations on the $\delta^{18}O$ value of moisture source vapor[36]. In summary, these factors at this time would allow cooling in the North Atlantic to result in atmospheric cooling throughout the circum-Arctic.

Because lakes in the MatSu are maintained by groundwater recharged by snowmelt (see Methods and Supplementary Fig. 1), it is most likely that the decreases in $\delta^{18}O_p$ during the YD reflects colder winter temperatures[8,9]. In contrast, summer-sensitive proxy evidence from our lakes builds upon existing studies that indicate relatively warm YD summer conditions[37–40], or inconsistent temperature trends lacking abrupt changes associated with the YD[12]. Evidence for warm summer conditions at our lakes during the YD includes high biological productivity and continuation of shallow water marl accumulation, minor changes in terrestrial vegetation, and evaporative enrichment of lake water $\delta^{18}O$ (see Supplementary Figs. 3 and 6). The best explanation to account for mild and relatively dry summer indicators, and the negative $\delta^{18}O$ excursions in both open and closed lakes during the YD, is significantly colder winters and relatively minor changes in summer[41].

Cooling during the YD is not obvious in most proxy records of ecosystem change, or in temperature sensitive glaciers and ground ice in the North Pacific-Arctic sector[3]. Rather, we suggest that seasonally cold temperatures during winter in Alaska occurred during the YD, similar in timing and structure with those in Greenland, without strongly affecting proxies sensitive to summer conditions, analogous to seasonally sensitive records from the North Atlantic region that also show winter cooling and relatively warm summers[41–43]. Enhanced seasonality during the deglacial period may have resulted from precession driven summer warming, which accelerated ice-sheet meltwater discharge and weakened NADW, which in turn favored sea-ice growth and greater continentality, and subsequently colder winters[43].

### The early Holocene atmospheric transition

During and following the YD termination, ca. 11.7 to 11.5ka, winter $\delta^{18}O_p$ and summer evaporation reach maximum values, suggesting a brief period of unprecedented warming during summer and winter (Figs. 3 and 4). Higher $\delta^{18}O_p$ values could also occur due to an increase in rainfall relative to snowfall and/or increased Bering Sea moisture sources related to melting sea ice, which are both also consistent with winter warming. Subsequently, as Bering Sea sea-ice coverage declined and Alaska's shoreline moved inland with rising sea levels, the resulting loss of continental area led to a more maritime climate which is documented by rising lake-levels during this period[44,45]. Climate models indicate that seasonal temperatures were sensitive to these changes in Alaska, with cooler summers in coastal areas and warmer winters in the interior that would have shifted the seasonal precipitation balance towards rainfall[46]. After ~11 ka, decreases in winter $\delta^{18}O_p$ (Fig. 3a) are interpreted to reflect increased contributions from southerly moisture sources, possibly in response to Bering Strait flooding[47]. Opening of the Bering Strait led to increased freshwater export from the Arctic Ocean into the North Atlantic known to weaken AMOC[48] as tropical Pacific Ocean temperatures were rising[4] (Fig. 2) and the AL was strengthening[46], which would have favored northward moisture transport. As discussed previously, this leads to lower $\delta^{18}O_p$ values in Alaska[16,17].

These hydrologic changes in Alaska promoted thermokarst[49], which increased intra-permafrost groundwater flow and hydrologic connectivity[50,51] and accelerated peatland expansion[52,53]. As respiration of organic matter in soils increases $pCO_2$ and lowers the $\delta^{13}C$ values of dissolved inorganic carbon in groundwater recharge, changes in this input to lakes can be recorded by the $\delta^{13}C$ of sedimentary organic matter. Thus, increasing surface runoff and groundwater recharge flux to lakes will impact carbon pools and their isotopic signatures by resulting in decreases in the $\delta^{13}C$ of aquatic plants and sedimentary organic matter (see Methods and Supplementary Figs. 8–10).

The April Fools Lake organic $\delta^{13}C$ record shows high values during the YD, with a significant decrease at the end of the YD-early Holocene transition (Fig. 3b). Observations from additional lake records across interior Alaska show similar trends during this period (Figs. 1, 3c and

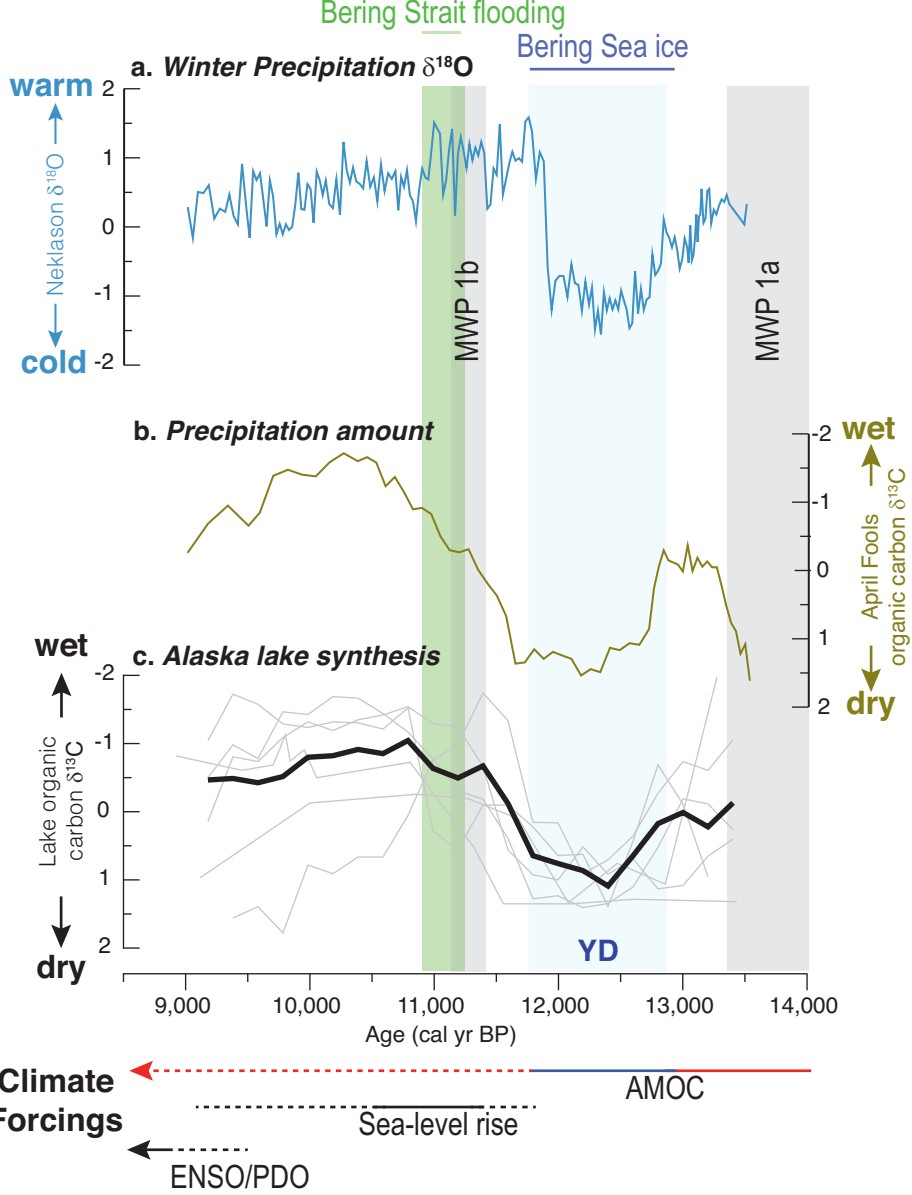

**Fig. 3 | Alaska temperature and precipitation changes 9 to 14ka. a** Winter temperature is inferred between 13.5 and 11.5 ka from North Pacific-Arctic $\delta^{18}O_p$ z-scores (Neklason Lake, blue) shown with **b**, MatSu precipitation trends inferred from lake sediment organic $\delta^{13}C$ (April Fools Lake, olive, note the inverted anomaly scale), and **c** interior Alaska precipitation trends inferred from a synthesis of sediment organic $\delta^{13}C$ z-scores (*n* = 6, black, note the inverted anomaly scale, individual records in light gray; see supplementary Fig. 8). Horizontal arrows summarize shifting climate forcings (solid strong, dotted weak, red warm, blue cool). YD Younger Dryas, MWP melt water pulse events, AMOC Atlantic Meridional Overturning Circulation, ENSO/PDO El Niño Southern Oscillation/Pacific Decadal Oscillation.

Supplementary Fig. 8). Significantly, carbon to nitrogen ratios of organic matter in these lakes remained relatively low and invariant, and organic matter content varied little, indicating predominantly aquatic plant carbon sources and little change in productivity that could otherwise impact $\delta^{13}C$ (Supplementary Fig. 9). This data supports the idea that precipitation increased regionally after the end of the YD, resulting in higher groundwater fluxes through newly forming permafrost-affected soils[54], creating similar patterns of $\delta^{13}C$ variations among a wide variety of lakes. During the YD, elevated $\delta^{13}C$ suggests drier conditions consistent with greater summer evaporation that is documented by elevated Finger Lake $\delta^{18}O_c$ (Fig. 4), though moisture remained sufficient to support lake formation. Declining $\delta^{13}C$ during the early Holocene corresponds with moisture rising to levels higher than previously recorded. In contrast, some later Holocene $\delta^{13}C$

differences among the lakes reflect unique lake development trajectories related to bathymetric setting during later stages of lake ontogeny and terrestrial vegetation change (see Supplementary Fig. 10).

### North Pacific controls on Alaska $\delta^{18}O_p$ during the Holocene

After the early Holocene transition, our records indicate relatively stable winter $\delta^{18}O_p$ and reduced summer evaporation effects, a previously recognized trend associated with AL intensification and increased supply of moisture from the subtropical Pacific Ocean[55] (Fig. 4). The period from ~10 to 4 ka was wetter than previously recorded and experienced low magnitude $\delta^{18}O_p$ variance (±0.5‰) around mean values similar to modern. These controls on $\delta^{18}O_p$ in Alaska during this period contrasts with that in the Greenland record, which is interpreted to reflect temperature sensitivity to AMOC[56]. Also,

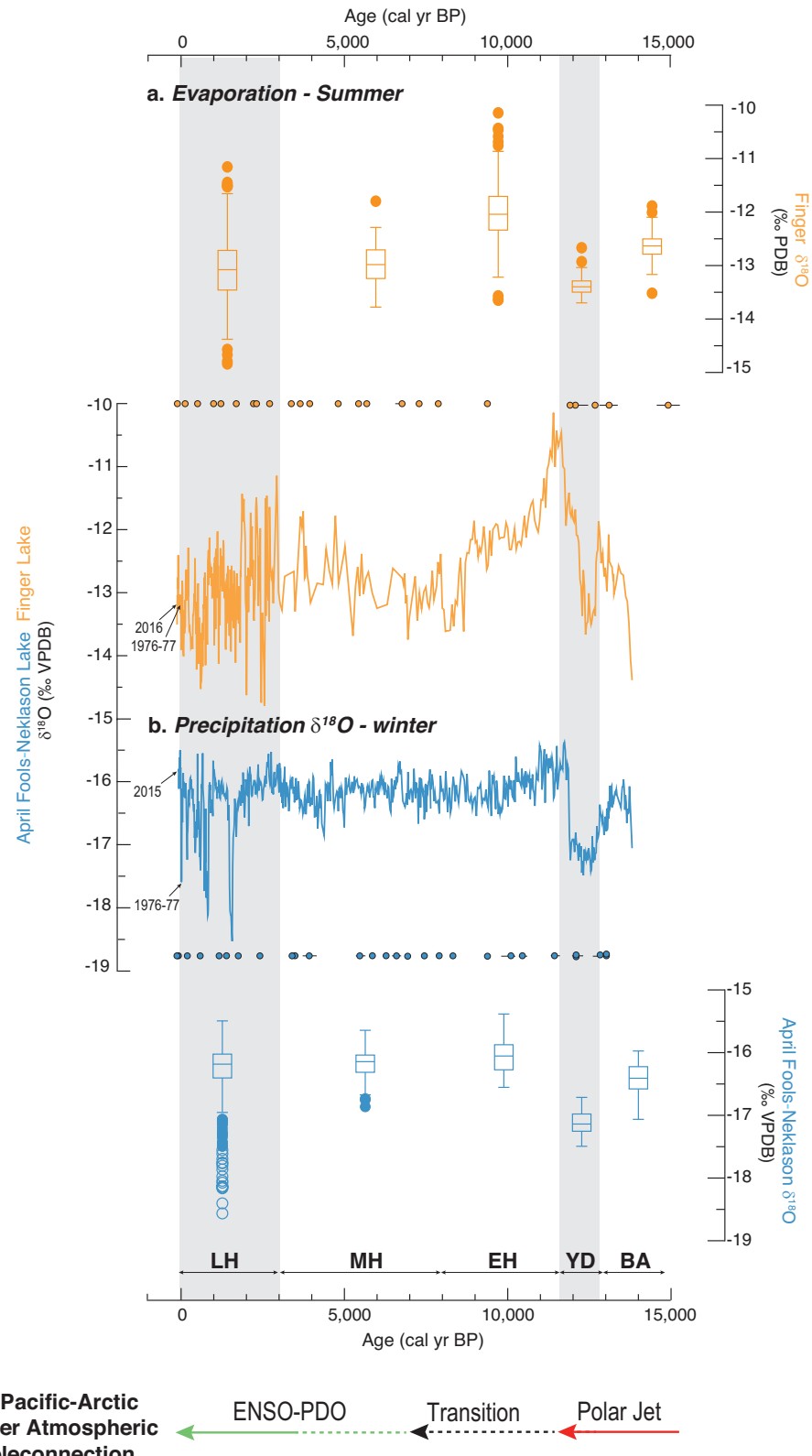

**Fig. 4 | Alaska lake records of winter precipitation δ¹⁸O and summer evaporation.** $\delta^{18}O$ records with box plots for time intervals identified on the horizontal axis (median, 1st and 3rd quartile, outliers) for **a**, Finger Lake (orange), closed with sensitivity to summer evaporation, and **b** April Fools-Neklason Lake (blue), open with sensitivity to winter precipitation $\delta^{18}O_p$. Colored circles above and below each $\delta^{18}O$ record indicate age points (1-sigma range if outside symbol size). Boxplots during the YD and late Holocene show relatively low $\delta^{18}O$ values in open and closed lakes (gray areas). Horizontal arrows summarize proposed shifts in winter atmospheric teleconnection. LH Late Holocene, YD Younger Dryas, BA Bølling Allerød.

unlike Greenland, Alaskan lake $\delta^{18}$O records show no anomalous climate signals at 8.2 ka in line with weakened connections to North Atlantic forcing (Fig. 4), or at 4.2 ka.

By the late Holocene, both the magnitude, and absolute decreases of winter $\delta^{18}O_p$ variations in our records intensified, with some events after ~2ka having values more negative than recorded during the YD (Fig. 4b). This winter trend is best explained by intensified meridional circulation patterns (in contrast to zonal flow) driven by ENSO/PDO[6,57], the frequency of which increased during the late Holocene (Fig. 2d) and as discussed results in significant $^{18}$O-depletion[16]. This trend is also consistent with increased variance and high magnitude negative $\delta^{18}$O excursions found in other records across the margin of western North America[4–7,58–62]. The long-term trend in $\delta^{18}$O in summer-sensitive Finger Lake, also shows higher magnitude variations and the lowest $\delta^{18}$O values of the entire record (Fig. 4a). However, the timing of many of the multi-decadal scale variations do not appear to coincide, which likely illustrates how open and closed lakes reflect different seasons of the same climate. Critically, summer climate reflects circulation patterns when the AL is not the dominant control and therefore differs from winter[16,17,21].

### The significance of winter climate shifts

Wintertime circulation patterns are well recognized in driving basin-wide patterns in land and ocean temperatures and precipitation, ocean productivity, and marine fish abundance[63,64]. This long-term context helps frame more recent disruptions of highly variable atmosphere-ocean systems that clearly undergoes shifts in drivers through time[65,66]. For example, our records document greater extremes in pre-historic baselines that help evaluate recent marine heat waves and associated sea bird and fish die-offs[67,68]. In the terrestrial realm, independent evaluations of long-term baselines in regional carbon dynamics, including biogeochemical changes associated with permafrost and hydrologic connectivity, may be compared in light of the seasonally complex and highly variable hydroclimatic history provided here[69,70].

Our Alaskan lake $\delta^{18}$O records, which provide seasonal information with solid chronological control, verify the existence of a YD signal transmitted to Alaska by the atmospheric during winter. This is a particularly important discovery in the North Pacific-Arctic as identifying this event and determining its linkage with the North Atlantic has previously been difficult, creating uncertainty about the interplay of Atlantic and Pacific teleconnections through time. We confirm severe YD winter cooling with modest reductions in moisture and mild summer conditions. This accounts for the absence of glacial moraine evidence for advances during the YD chron[37] and conflicting proxy records that largely reflect summer conditions in terrestrial (i.e., pollen), and marine environments (e.g., Gulf of Alaska alkenones which also reflect warm seasons)[71].

Following deglaciation, major sea-level rise, and the opening of the Bering Strait, we document a weakened North Atlantic teleconnection and greater control originating from North Pacific subtropics, which continues to have global influence today. This shift may represent the effects of changing orbital precession on expanding latitudinal ENSO gradients[5], which are best assessed by winter proxy records within the North Pacific region and are currently scarce[72]. Moving forward, thorough examinations of shifting global climate influences over time will clearly be strengthened by well-dated, high-resolution terrestrial paleoclimatic data sensitive to winter.

Our records illustrate the role of seasonality and circulation patterns with regard to understanding abrupt climate change and isotope systematics. The data show that similar decreases in lake sediment records of winter precipitation $\delta^{18}$O can be forced by different mechanisms. Such results provide constraints for isotope enabled climate model simulations that commonly miss the dual role of seasonal moisture and temperature in high latitudes[73]. The shifting dominance between Atlantic and Pacific influences on Arctic environments identified here emphasizes the need to better represent Pacific dynamics in climate models, particularly given their broad hydrologic and ecological impacts. Our data documents the major roles of seasonality and shifting atmospheric patterns into our conceptual understanding of linkages between regional and global scale climate oscillations and rapid climate change, past, ongoing, and future.

## Methods

### Study lakes, hydrology and climate

All study lakes are biologically productive, fed by alkaline, calcium-rich groundwater. This contributes to the formation of shoreline marl benches, which are composed of well-preserved authigenic carbonate sediments (see Supplementary Table 1 and Supplementary Table 2). April Fools Lake (informal name: 61.49°N, 150.0 °W, 41 m a.s.l.) is our western-most study lake, near the terminus of the regional groundwater flowpath[15]. The lake is maintained by several artesian springs along the northeast shoreline and has a surface outlet. Neklason Lake (61.63°N, 149.3°W, 128 m a.s.l.) and Finger Lake (61.60°N, 149.3°W, 103 m a.s.l) are to the east, nearer to the upland groundwater recharge area. Neklason Lake has a surface outlet and is maintained by a surface stream from an upstream lake sourced by springs. Finger Lake has no surface inflow or outflow and is maintained by groundwater.

We developed a lake water isotope dataset for April Fools Lake and compiled existing water isotope data for the MatSu Valley including precipitation, springs, groundwater, Neklason Lake, Finger Lake, and other lakes[74] (see Supplementary Fig. 1). The data show variations in the $\delta^{18}$O of groundwater by 2–3‰ that reflect an integrated multi-year $\delta^{18}$O signal of precipitation, which varies seasonally up to ~9‰[15,17]. Regional trends in lake water $\delta^2$H and $\delta^{18}$O defines a local evaporation line (LEL) with a slope of ~4. Hydrologically open lakes like April Fools Lake and Neklason Lake have relatively low $\delta^2$H and $\delta^{18}$O values that indicate they are good representatives of regional groundwater $\delta^{18}$O values and thus $\delta^{18}$O of precipitation. Hydrologically closed lakes, such as Finger Lake, have relatively high $\delta^2$H and $\delta^{18}$O values due to summer evaporation effects which are clearly distinguishable from $\delta^{18}$O of precipitation and groundwater[15,17].

The climate of the MatSu Valley is a transition zone between maritime and continental climates with an average seasonal temperature range between −12 and 21 °C. Average total precipitation is 38.8 cm in the valley increasing with elevation to ~90 cm in the local headwaters of the Talkeetna Mountain foothills (~2000 m)[15] and to substantially higher amounts at elevations of ~4000 m. Within the contributing alpine watersheds that recharge the groundwater aquifer, the majority of annual precipitation is snowfall between October and April. This results in groundwater fed MatSu lake-water $\delta^{18}$O values strongly weighted towards controls by winter atmospheric circulation, moisture origin, and transport pathways[17], which proxy evidence also show during the Holocene[16].

### Lake sediment lithology and carbonate isotopes

Sediment cores were retrieved from lake ice, or a securely anchored floating platform, with a modified Livingstone piston corer (Supplementary Table 2). Cores with an undisturbed sediment water interface were obtained using a customized polycarbonate tube fit with a piston and extruded in the field at 0.5-cm increments. At the coring sites of our study lakes, the water environments are clear and quiet (Secchi disk depths >3 m), which promotes the precipitation of calcium carbonate within surface waters by photosynthetic alterations of bicarbonate equilibrium (e.g., bio-induced carbonate precipitation)[75]. The geomorphic and hydrologic configurations of the small low-relief watersheds (<50 m elevation) within wetlands strongly limits

watershed transport of allochthonous (i.e., detrital) carbonate sediment. Regional aeolian loess is mainly composed of quartz, plagioclase, mica, amphibole and chlorite, and is low in carbonates[76].

We developed a suite of sedimentary analyses that document changes in paleolimnology that help to evaluate controls on the sediment carbonate isotope data. They include dry bulk density, magnetic susceptibility, and percent organic matter and carbonate by loss on ignition (see Supplementary methods and Supplementary Fig. 3). The similar lithology for all of the cores overlying basal deglacial deposits is dominated by two authigenic components, olive to light gray bedded marl or dark brown marly gyttja to green brown gyttja. The only exceptions are two volcanic tephra that are easily identified, either visually by abrupt changes in color, density and texture and by sharp increase in magnetic susceptibility.

All cores have similar light and dark sediment bedded structure which reflect changes in the relative proportions of preserved organic carbon and calcium carbonate. The residual mineral content is typically <10% of sediment weight and comprised of mixtures of biogenic silica and non-carbonate minerals. The carbonate contents of shallow water sediments from April Fools Lake and Neklason Lake is often high (~80 to 90% carbonate; see Supplementary Fig. 3a and Supplementary Fig. 3c and d) and dominated by micrite. Finger Lake shallow water sediments are comprised of pure micrite in the basal unit, marly to highly organic gyttja in the middle unit, and marly gyttja in upper unit (see Supplementary Fig. 3e). April Fools Lake sediments from water depths below the thermocline are predominantly composed of marly gyttja and gyttja due to reduced carbonate preservation caused by dissolution within the water column; our analyses of a core from this region (A15) focused on organic carbon and nitrogen (see Fig. 3 and Supplementary Fig 3b).

An authigenic origin of the bulk sediment carbonates was verified by good agreement between surface sediment oxygen isotope composition and that calculated for isotopic equilibrium between carbonate and lake water (see Supplementary methods and Supplementary Table 4 and Supplementary Fig. 4). Samples for carbonate isotope analyses were taken at contiguous 0.5-to-1-cm sample increments, although in some core sections carbonate contents were found to be too low for analysis (see Supplementary Fig. 3). To facilitate separation of bulk authigenic carbonates for isotope analyses from other larger carbonate particles of biological origin (e.g., ostracode valves, gastropods, and bivalve shells and their fragments) and allochthonous processes (i.e., terrestrial plant macrofossils), samples were wet sieved through nested screens (250, 125, 63, 32 μm), collected separately and freeze-dried. The finest fraction (<32 μm) was analyzed for most samples. For a few samples (<1%), the <32 μm size was combined with the 32 to 63 μm sizes to obtain sufficient mass for analysis because insignificant $\delta^{18}O$ difference was found among smaller grain sizes (see Supplementary methods and Supplementary Table 3).

### Age data and age-depth models
Sediment core age data are based on [210]Pb analyses, two prominent tephras identified as sourced from Mount Hayes, and AMS [14]C measurements on macrofossils of terrestrial origin (see Supplementary methods and Supplementary Figs. 3–5 and Supplementary Tables 5–7). Bayesian age-depth modeling provide average 95% certainty bounds of <± 200 years among all of the records (see Supplementary Table 6). Carbonate sedimentation rates provide sub-decadal to multi-decadal resolution from present until ~8 ka and multi-decadal to centennial scale resolution between ~8 and 14.5 ka. For the $\delta^{13}C$ records in Fig. 3, we employed the available age models for Arolik Lake and Dune Lake. The existing radiocarbon-based age models for Birch Lake, Little Harding Lake, and Eightmile Lake were refined with additional radiocarbon ages and recalibrated using the IntCal20 radiocarbon calibration curve.

The April Fools Lake core A15 $\delta^{18}O$ chronology is based on [210]Pb, two tephra layers, and 5 radiocarbon ages of terrestrial macrofossils (see Supplementary methods and Supplementary Fig. 3a and Supplementary Fig. 5a). The Neklason Lake core B16 chronology is based on 18 radiocarbon ages of terrestrial macrofossils (see Supplementary Figs. 3c and 5b). A second core from Neklason (A16) reproduced similar $\delta^{18}O$ variations across the YD in the lower section (see Supplementary Fig. 5b). To optimize the high-resolution portions of the April Fools Lake and Neklason Lake records, their $\delta^{18}O$ stratigraphies were merged into a single high-resolution stable isotope record (see Supplementary Fig. 7). The independent chronologies were spliced together at 7.6 ka with no scaling or adjustment of the isotope scales and verified by the stratigraphically ordered ages. Two cores obtained from Finger Lake were stratigraphically correlated to develop a composite stratigraphy (core A16/23) and the chronology is based on [210]Pb, two tephra layers and 22 radiocarbon ages of terrestrial material (see Supplementary Figs. 3e and 5c). The Bayesian age-depth models for each of our lake $\delta^{18}O$ record shows correlation during the YD to Greenland within 95% confidence limits.

### Lake sediment carbon and nitrogen of organic matter
The percent organic content and its C/N ratios can be used to assess factors that may influence the $\delta^{13}C$ of organic matter in lake sediments internal and external to lakes (see Supplementary methods). The records indicate low and relatively invariable C/N between 9 and 13 ka, suggesting an aquatic source material throughout the period (see Supplementary Fig. 8 and Supplementary Fig. 9). Minor variations in organic carbon content suggests limited changes in productivity across the interval. If driven by lake productivity, the sediment decreases in $\delta^{13}C$ would infer a chronic productivity decline, which is not observed in the records. Atmospheric $pCO_2$ and its $\delta^{13}C$ were relatively constant during this period, suggestion minimal influence from atmospheric exchange processes. While water temperature affects the solubility of $CO_2$, if the observed $\delta^{13}C$ decreases were due to temperature change, this would imply a significant decline in temperature during this period, which is not supported by paleoenvironmental data. Thus, we conclude that the $\delta^{13}C$ decrease during the interval between 9 and 13ka most likely reflects a major regional increase in hydrologic input to lakes[50].

### Data availability
The data generated by this study have been deposited in the U.S. National Oceanic and Atmospheric Administration (NOAA) National Center for Environmental Information (NCEI) under the accession code https://doi.org/10.25921/jvs7-8f93.

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

## Acknowledgements

We thank Carson Baughman, Matt McMillan, Eva Stephani and Ben Gaglioti for their valued assistance in the field, Million Hailemichael and the Idaho State University Stable Isotope Laboratory for analyses, and Nancy Bigelow for providing the pollen analyses. This research was supported by the USGS through the Ecosystems Land Change Science program. Any use of trade, firm, or product names is for descriptive purposes only and does not imply endorsement by the U.S. Government.

## Author contributions

L.A. and B.F. conceived and designed the study and performed the field work, analysis, and data compilation with assistance by W.B.B. L.A. and B.F. interpreted the data with input from W.B.B. L.A. drafted the paper and with B.F. generated the final manuscript.

## Competing interests

The authors declare no competing interests.
