## [Transparent Peer Review file · Nature Communications]

Shifting Winter Atmospheric Teleconnections to the North Pacific reconcile Younger-Dryas and Holocene $\delta^{18}\text{O}$ signals

Corresponding Author: Dr Bruce Finney

Version 0:

Reviewer comments:

Reviewer #1

(Remarks to the Author)

In this manuscript, the authors present high-resolution oxygen isotope records from several lakes in Alaska to investigate climate oscillations during the last deglaciation and compare with the late Holocene. They show that the YD mostly represents winter cooling and is closely tied to Atlantic Ocean influences, supporting the existence of strong connections between North Pacific and North Atlantic during the last deglaciation. On the other hand, rapid climate oscillations during the late Holocene were mostly triggered by Pacific Ocean influences, showing no evidence of influence from North Atlantic. They also discuss the implications of the study for recent and ongoing climate changes in North Pacific region.

This is an important study addressing a long-standing paleoclimate question, not only about the connection between North Atlantic and North Pacific, but also the variable nature of this connection. The isotope records presented are valuable and high quality. The interpretations are balanced and robust, supported by the data and arguments. The study design of using lakes with contrasting hydrology and the interpretations of organic d^{13}C records for groundwater flux and moisture conditions are creative and novel. The manuscript is well organized.

My only major comment is about presentations of dating controls. It is well known that lacustrine sediment records are very difficult to data from lakes in Alaska and Arctic in general, as suitable terrestrial plant materials are hard to find. The authors should be commended for such a large set of radiocarbon dates from these study lakes as presented in SI Table 1. That being said, dating and chronology are critical for the interpretations and conclusions presented in this study, so I suggest that the authors show explicitly the dating control points on the records shown in figures (comment below as well) and discuss any uncertainties accordingly. That way, readers would have a visual appreciation of the robustness of chronology.

Specific comments:

Abstract: mention lake records to make the abstract more informative

L64: change "guided" to "modulated"?

L82-83: Refer Fig. 2b for Neklason Lake record, and d^{18}O from April Fools Lake is not in Fig. 2, but in Fig. 4. Also, should refer Fig. 2a after mentioning Finger Lake here.

L89: age models: need cite Suppl Figures?

L93: Change superscript 15 to "ref. 15"

L96: "from ~4 to 8 C". Negative signs have been taken care of by "decline" in the sentence, so not needed here.

L95-97: unclear. Temperature decline is inferred/estimated from d^{18}O change based on calcite-water fractionation, but what is the role of SS temperature variation here. The former was used for temperature inference/estimate, while the latter is for interpretations. Do I miss anything here? Clarity.

L98: perhaps compare d^{18}O values of winter precipitation/groundwater at the sites and lake-water d^{18}O values calculated from calcite d^{18}O measurements using calcite-water d^{18}O fractionation factor

L101: change to "...sensitivity; however,"

L111-115: add dating control points near curves on figures to support/refute the YD timing discussion here

L115-116: good point! The fact of calcite precipitation suggests that summer water temperature is high enough.

L121: "meteoric precipitation" to avoid possible confusion with calcite precipitation?

L134-143: this paragraph needs to refer to Fig. 3, perhaps different panels according to various descriptions, including to April Fools Lake (Fig. 3b) and synthesis records (Fig. 3c). That would greatly increase clarity and readability.

L155-157: refer Fig. 3 for MWP mention

L171-172: Be mindful here. Finger and especially Neklason records have different sampling resolution, so potentially artifact for variability/magnitude. But indeed, April Fools record is a strong proof that variability/magnitude shift around 2 ka.

L180-183: very good statements/examples about implications of this study. Well done.

Figure 2: add dating control points on the figure, perhaps on the top with different colors (such as downward triangles) for Finger and Neklason records, even for composite records.

Fig. 2c: vertical axis label: change to superscript in m²

L235: change to "Greenland ice-core"

L244: Fig. 3 caption: indicate the number of organic d¹³C records (n=)?

L251: a stray bracket ")"

L258-259: Livingstone corer can hardly recover sediment-water interface. Either you use a plastic piston corer, or delete the mention as it is not essential for the study.

L260: we use "endogenic" to describe carbonate, not d¹⁸O carbonate, right? rephrase.

L290: an extra ""

Supplementary Information:

L52, 59, 67: Bayesian

SI Fig. 3b. vertical axis label "d¹³C"

SI Fig. 4: n=6 for d¹³C records need to mention in the main text/figure

SI Figs. 5 and 6 are apparently not mentioned in the text, It would be good to do so, when describing the nature/source of organic matter for d¹³C interpretations.

SI Table 1:

3rd column heading: 14C Date (not age here, but "Calibrated Age")

4th column heading: "...Date (14C yr BP)", not "years"

L116: "informal name". Hopefully next time when the lake was cored again on July 4th, it wont be named "Independence Lake":))

L120: "at two locations"

L292: why don't make it a formal SI Table and formatted as such? The table heading needs proof read, as they are incomplete/inaccurate: such as "Horizons dated", basal age need "Cal yr BP"? Sampling resolution: "cm" can be moved to heading (should cm be converted to years, as you have age model already?)

Reviewed by Professor Zicheng Yu

Reviewer #2

(Remarks to the Author)

Review on Anderson et al. submissions to Nature Communications

This paper presents new isotope results for lakes in Alaska that span the Younger Dryas to Late Holocene, providing interesting high-resolution records that are presented nicely in the figures and may eventually advance our knowledge about the Pacific-Atlantic/Arctic teleconnections. As detailed below, while the topic and dataset are promising, the manuscript requires a substantial revision and reorganization to clarify its framing, improve methodological transparency, and provide a more balanced interpretation of the results.

1. General Comments

1) Chronology and lake reservoir corrections

The age-depth model is central to the paper's conclusions. However, it lacks sufficient justification and statistical rigor during chronology establishments. I do agree with the idea using the tephra layers with absolute dates to correct the lake reservoir effects. However, I also believe that more details are needed for better clarity:

a. How the tephrochronology was established for those cores? The authors claim that the tephra correlation is established "Based on similarities" (Line 217 in SI). However, it's not very clear to the readers what does "similarities" represent. Have you run the geochemical analysis on tephra shards? Which is the routine way for tephra layer correlations. I would suggest the authors provide more robust information for tephrochronology correlation, either other evidence to clearly show the "similarity" or geochemical analyses results on tephra shards. This will directly influence the reliability of lake reservoir effects estimation and following age-depth modelling for lake April Fools.

b. Reasons for the rejection of some radiocarbon dates should be given or detailed. This study presents a massive amount of radiocarbon dating on those cores (which is good) but they also rejected some of those radiocarbon dates but give very little discussion why they did this. Taking core April Fools A15 as an example, the authors rejected more than half of the radiocarbon dates (including the lake reservoir corrected aquatic herb-based dates), but the reason were not given/detailed.

2) On the interpretations of stable oxygen isotope ($\delta^{18}\text{O}$) of the carbonates.

a. Why is your proposed driver the most likely? Lake carbonates $\delta^{18}\text{O}$ signals are influenced by multiple factors, including temperature, evaporation, seasonality of temperature/precipitation, and hydrological conditions. While the authors try to classify these three lakes into two groups, namely Precipitation-type and Moisture-type (Lines 53-62), to simplify the interpretation of $\delta^{18}\text{O}$ signals, the manuscript does not convincingly justify why those particular factors are the dominant control as claimed (even without reference supporting their classification)?

b. Problem with mixing carbonate types. The manuscript does not clarify the type of carbonates being analysed. From the method section, it seems that the fine fraction ($<32\ \mu\text{m}$) was used for isotopic works. However, this fine fraction seems to be a mixture to me, and this carbonate source ambiguity raises concerns about the geochemical integrity of the proxy signal. For example, what is the proportion of biogenic, inorganic, or even detrital carbonate in the finest fraction ($<32\ \mu\text{m}$)? Are there sedimentary or geochemical indicators that confirm the origin of the carbonates?

c. There is currently no discussion of possible isotopic offsets between different carbonate fractions. This is a crucial omission, as differences between, for example, biologically precipitated carbonate and inorganic calcite can introduce systematic biases. Is there any isotopic offset between different grain-sized fractions, between fractions $<32\ \mu\text{m}$ and $32\text{-}63\ \mu\text{m}$? Change in the carbonate source can influence the isotopic signals of the carbonates, which is independent of climate change.

I would recommend:

- i. Consider running modern calibration studies on modern water or carbonates samples to enhance the interpretation of the isotopic signals of carbonates (e.g., comparing $\delta^{18}\text{O}$ of precipitation, lake water, and carbonates with climate parameters).
- ii. If only bulk carbonates were analysed, the potential impact of this choice on the paleoclimate interpretation should be clearly discussed and acknowledged as a limitation.

3) On the interpretations of bulk organic stable carbon isotope ($\delta^{13}\text{C}_{\text{org}}$)

The authors interpret the $\delta^{13}\text{C}$ values of bulk organic matter primarily as a proxy for precipitation variability. However, this interpretation appears overstated and does not adequately consider alternative influences.

The C/N ratios reported (generally <15) suggest that the primary source of lake organic matter is aquatic algae. This implies that $\delta^{13}\text{C}$ variations likely reflect lake-internal productivity processes rather than external hydrological inputs, at least not only external processes. Aquatic algae assimilate dissolved CO_2 , and the isotopic signature of that carbon pool is influenced by lake water pCO_2 , which itself is affected by temperature, biological activity, and CO_2 exchange with the atmosphere. These internal carbon cycle dynamics can significantly modulate the $\delta^{13}\text{C}$ signal, making it problematic to directly attribute isotopic shifts to precipitation changes alone.

In its current form, the manuscript risks overinterpreting $\delta^{13}\text{C}$ as a direct climate signal, and I recommend the authors reconsider this conclusion and broaden their interpretive framework.

4) Mismatch between the paper's stated goals and its actual contributions.

While the manuscript presents interesting datasets from multiple lake records, I find that it ultimately does not achieve the stated goals outlined in the introduction or title. The authors claim to resolve key questions about climate connections between the Pacific and Arctic/Atlantic, yet the evidence presented—particularly regarding the $\delta^{13}\text{C}$ and $\delta^{18}\text{O}$ interpretations—remains ambiguous and open to alternative explanations that are not adequately explored. This ambiguity in the climatic significance of the proxy signals, combined with uneven methodological transparency (e.g., tephrochronology, lake reservoir corrections, unseparated carbonate fractions), undermines the strength of the conclusions. As a result, the study does not convincingly advance our understanding of teleconnections between Pacific and Arctic/Atlantic in the way it aspires to.

More supporting evidence is needed to back up that the lakes reflect winter conditions (references are not given in several places), and the authors often argue that changes reflect temperature, without explaining why precipitation source and atmospheric circulation is discounted. Furthermore, the lake records show very different variations during the late Holocene, but rather than acknowledge this and explain why this might be (and give a reasonable explanation of why they still have a climate signal) the authors discuss the high variability as being significant.

Finally, at some points in the paper the reader is left behind because the links between ideas are not explained fully. In parts the text requires reorganisation and rewriting, as comparisons with other records or supporting evidence is spread over several paragraphs.

2. Moderate comments

Line 53-62 – there is just one reference provided for the interpretations in this paragraph, do the interpretations all come from here? Are there modern observations or data to support this? References or data are needed particularly to support the interpretation of these being open/closed basins and the water residence times, which are the basis for the interpretation about seasonality.

Line 67-68 – ‘Contemporary ^{18}O precipitation reflects seasonal air temperatures and shifting moisture sources in addition to topographic rain-out effects’. This sentence sums up some of my concerns later in the paper where changes in the isotopic records are interpreted in certain ways, such as temperature, without considering the other possible causes given here.

Line 77 – ‘resolving past seasonal changes’. In the next section you explain why the lakes reflect winter changes but that is not explained well here. I think add to the end of this section more detail and references supporting how you know it reflects winter season changes.

Line 81-106 – These two paragraphs could be reorganised – it is quite confusing currently having comparisons with existing records spread over two paragraphs, and the statement about the YD being a winter phenomenon here only makes sense with the evidence shown in the following paragraph

Line 98-100 – ‘Our YD $\delta^{18}\text{O}$ carbonate records largely reflect winter season cooling’. How do you know it reflects cooling, could winter season circulation changes also have had an effect?

Line 102 – ‘Our new data substantiate that the YD in Alaska was most sensitive to wintertime cooling,’ The phrasing of this could be improved as currently the YD is sensitive, rather than the YD being a time period and the climate or winter season climate being sensitive. The references given later for the YD summer being warm should also be put here (currently down at line 112) as should the evidence from the lakes themselves (high productivity and marl accumulation) currently at line 115. This would strengthen the argument that the record is reflecting the winter rather than summer. Consider also my point for line 98 about the interpretation of this as cooling rather than atmospheric circulation, how do you know this?

Line 126-129 – references are needed here to support this interpretation

Line 143 - I am not sure if moisture shifts is the right term here for what you describe – a shift would suggest to me a change in the distribution of moisture across the area (so maybe some areas having more and others less). But as they are all changing together, are they not just experiencing enhanced moisture?

Line 146-148 – ‘both lake types indicate the warmest year-round temperatures and possibly the greatest aridity of the records’. In the Main section above when describing the causes of oxygen isotope changes the atmospheric circulation paths are described – how do you know therefore that the oxygen isotope changes are due to temperature changes and not changes in atmospheric circulation? Both types of lake are influenced by the precipitation, so this seems more likely to be causing shared changes?

Line 148 – ‘However, with winter experiencing the most significant warming, high $\delta^{18}\text{O}$...’ Is this based on the cold YD temperatures? How do you know this?

Line 146-159 – In this paragraph I feel like a few steps are missed in the explanation – I think you explicitly need to mention that it is thought that freshwater export may have weakened the AMOC (if this is what the references here say) and say how that would have caused warming/zonal circulation in Alaska. And was it the Bering Strait flooding or AMOC changes that you think caused the changes in Alaska? If you don't know say more clearly that there are a few potential explanations, or different causes at different times (as suggested by figure 3).

Line 163-164 – ‘This contrasts with Greenland, which experienced continuous temperature sensitivity to AMOC’. A reference is needed for this.

Paragraph starting 174 – here you change the interpretation from oxygen isotope variations being caused by temperature (during the deglacial) to atmospheric circulation. This interpretation is given very certainly, but without many references or explanation of why this change would occur. Furthermore, why are the records all showing different patterns if they reflect atmospheric circulation that would be the same across the area?

Line 180 – ‘For example, our records document greater extremes in pre-historic baselines that help evaluate recent marine heat waves and associated sea bird, and fish die-offs.’ This is a rather general statement – do you mean greater extremes than the 20th century variations? Greater using which variable? If you are able to say this then it should be supported by an example of how your results can be used to evaluate these impacts, or at least how modern observations of oxygen isotope values compare with the late Holocene range.

Line 197-199 – the links with precession and global changes should probably be introduced before the conclusion section. Without more explanation I don’t understand what is meant by this sentence, or how you have come to this conclusion. This is also discussing the late Holocene changes, which as I mention above are not consistent between your records, which I think is problematic for then making conclusions about the late Holocene climate.

Line 206-207 – ‘records show Late Holocene climate shifts comparable to those of the YD’ – you interpret that the isotope changes during the YD are caused by temperature, and during the late Holocene by circulation changes. So I am not sure you can then say they are comparable – it implies YD-scale temperature fluctuations were occurring in the late Holocene.

3. Minor Comments

Line 3, the title is concise and readable, but it does not fully capture the scope or key findings of the study.

Line 39 – ‘inform abrupt climate change’ – the word inform doesn’t fit here, perhaps ‘inform our understanding of abrupt...’ would work better

Line 52 – ‘unresolved timing’, this could be better phrased as ‘uncertain chronologies’. Unresolved timing requires more detail about what the timing is related to.

Line 66 – ‘oriented by coastal topographic barriers that were continental ice sheets during past glacial periods’, this phrasing is confusing. Do you mean the topographic barriers were shaped by the ice sheets? Or the barriers are currently topographic but there were ice sheet barriers in the past?

Line 73 and 74 – The use of the word ‘reflect’ here I think is unusual but perhaps ok. I would consider ‘cause’ instead

Line 74-76 – references needed here

Line 107 – add ‘timing and nature’ or something similar, as most of the discussion here is about the character of the YD rather than the timing.

Line 123 – ‘with a strong consensus for there having been an Early’

Line 210-211 – reference needed

SI Lines 292-293, Lake water pH or conductivity are needed. Lake water pH or conductivity measurements should be presented as geographical setting or background information. It can be useful for the interpretation of the carbonates $\delta^{18}\text{O}$ signals as disequilibrium offset increases at high alkalinity.

Reviewer #3

(Remarks to the Author)

Reviewer #4

(Remarks to the Author)

This is a nice paper that presents three new detailed carbonate $\delta^{18}\text{O}$ lake sediment records from Alaska. They records are of high quality and important for understanding regional paleoclimatic variations, and their potential links to abrupt climate changes in the North Atlantic ocean over the deglacial period.

The major comment I have for this paper is that, while great records and important, I don’t see what new discoveries were made about climate dynamics in the North Pacific. Some of the statements about the variability in the Holocene (after the more clear B/A and YD intervals) were not fully reasoned out, and their differences to the Mt. Logan ice core were not

explained. I can certainly see why insolation might be a control, as suggested, but I don't see the reasoning that supports such an interpretation, or why the Late Holocene $\delta^{18}\text{O}$ record should be interpreted differently than earlier periods.

In summary, this is a solid regional-level characterization of south central Alaskan Holocene paleoclimate from lake sediment records. The time series are of high quality and will be an important contribution to our understanding of regional paleoclimate. But the manuscript doesn't yet make a convincing case on the hemispheric or global forcings of the paleoclimatic changes, or why they are significant for our understanding of north Pacific climate dynamics.

Minor points

I don't think the authors need to redefine open and closed basin lakes as "precipitation-type" and "moisture-type". Just the description of open and closed and their relative sensitivities to precipitation $\delta^{18}\text{O}$ would suffice.

I presumed in the introduction that the authors would come back to the Pacific climate modes and show some linkages between proxies for those and the lake sediment records. The lake sediment oxygen isotope records were only qualitatively linked to these Pacific Ocean climate modes, without any conclusive evidence of them being important forcings. The authors can describe the source of the carbonate in the lakes up front. Are they measuring authigenic calcite, shells, or something else?

In line 174, the authors did not provide their evidence that late Holocene oxygen isotope values would have a different interpretation than during the deglacial period. Further, Rayleigh distillation is happening in all cases across all time intervals, and therefore is not unique to the Late Holocene, so further evidence needs to be provided to support the authors reasoning. Similarly, the manuscript would benefit (in line 198) a better description as to why orbital precession is now an important control over the Holocene. Is this backed up by model runs? And are the authors referring to local summer, winter, or other season for insolation?

In figure 2 the authors show the Mount Logan ice core record, which spans the entire time interval covered by the lake sediment records. So, what do the new lake sediment records provide for our understanding of deglaciation paleo climate that was not apparent in the Mount Logan ice core record? There are some obvious differences, with the Mt. Logan $\delta^{18}\text{O}$ record having a relatively non-trending Holocene, whereas the lake records show more change. What would explain the discrepancy?

Version 1:

Reviewer comments:

Reviewer #1

(Remarks to the Author)

The authors have carefully addressed my concerns on the earlier version of the manuscript and have significantly revised and improved the manuscript. I have no major comments.

I have some editorial suggestions:

Figure 3: Axis label "Age" should be "Age (cal yr BP)"?

Ref. 51 and 83 are the same reference.

Supplementary file:

Table S1: Upper cases and lower cases are inconsistent. For example, "Temperature range", "Specific Conductance", "Dissolved oxygen", etc., etc. (similar U/L cases issues with other tables as well)

Table S5: Change "210Pb Age AD" to "210Pb Year AD" or equivalence (2015.2 is not age (yr old), but date)

Table S6: "Dated horizons"?

Fig. S6: Pollen concentration needs unit

Reviewer #2

(Remarks to the Author)

The manuscript has been improved substantially by the changes made by the authors and it reads better than it did previously. The revision has increased the transparency of the methodology and strengthened the justification for its robustness. The findings related to the YD and Early Holocene are convincing and interesting, and well presented in the figures. The study is strengthened by the additional analyses or reorganization of the manuscript, and the work has the potential to be impactful.

That said, I am remain unconvinced about the significance of the variations for the late Holocene, which is a section of the manuscript that either requires better justification or for the conclusions to be altered.

Another concern is about the organization and narrative flow of the revision. The overall organization of the manuscript is still challenging to follow, especially for readers not familiar with the study area. While the manuscript has been improved, it

still requires some sentences or paragraphs to be rephrased for clarity or to provide more explicit explanation of the points that are being made.

Major point:

1. Although the authors have substantially revised the structure and flow in response to my previous comments, I remain concerned that the manuscript's organization is still not optimal. The introduction does not follow the general structure expected in a Nature Communication manuscript, which should clearly present the significance of the topic and the central knowledge gap. Given the text constraints, there may not be space for an extensive review of prior work, but it remains unclear what the main knowledge gap is and how this study addresses it. I recommend a sharper framing and restructuring of the introduction to highlight both the novelty of the work and its broader significance. For example, in Lines 35–48, the introduction begins by emphasizing the novelty of the research and technical details, which is not an effective way to start.

2. Following the Introduction, there is a section titled 'Conceptual Framework.' Its purpose is not entirely clear, and the content appears to be more introductory than what is currently presented in the Introduction. I recommend merging the 'Introduction' and 'Conceptual Framework' into a single, stronger Introduction, unless the journal requires them to remain separate.

3. In the previous review I questioned why the records differ for the late Holocene section if they are reflecting regional climate and atmospheric circulation, and this has not been explained. I don't think this has been addressed by inclusion of the ranges in figure 4 or the additional conceptual framework (you state at L105 'Within Alaskan records, differences in timing and magnitude of $\delta^{18}O$ changes imply variability in both temperature and precipitation resulting from atmospheric circulation patterns' – but across a relatively small region such as this changes in temperature and precipitation would be the same surely?). I cannot find anywhere else this has been discussed. Perhaps a comparison of the records showing just the late Holocene section in the supplementary file would show that they co-vary but have different magnitudes, it is hard to see if this is the case from the figure, which could be explained by local factors. Otherwise I don't see how you can conclude that the lake records are reflecting late Holocene changes in atmospheric circulation (as you do at line 220) as this would alter climate in a similar way at all the lakes.

4. The revisions have introduced valuable new text, data, and figures (primarily in the Supporting Information), which strengthen the manuscript scientifically. At the same time, this restructuring has created some clarity issues that were not present in the original submission. I would encourage the authors to emphasize the central findings more prominently in the main text and streamline the methodological details into the Supporting Information, while ensuring a clear and logical flow. This adjustment would help the manuscript become more concise and accessible to a broad readership, and ultimately enhance its impact and visibility.

5. Results and discussion structure. The results section currently contains extensive commentary that reads more like discussion (For example, Lines 126-134). Conversely, the discussion section is largely a continuation of further discussion. I would recommend merge results and discussion into a single, well-structured section.

Moderate/minor points:

Lines 26-28 – this is not a clear sentence, consider rephrasing

Lines 52-53 – the c and p on the $\delta^{18}O$ could be explained using brackets here

Line 64-67 - this is not a clear sentence, consider rephrasing

Line 77 – 'associated with intensified' – often words like an, or the, are missing through the main text. Please check this here and elsewhere.

Line 87-89 – references needed for the previous proxy studies

Line 90 – 'would imply unrealistic higher temperatures' is it unrealistic in magnitude? or just that you expect the LIA to be colder?

Line 98, I don't think "YD is the most recent stadial period"

Line 104 – 'that are not observed in the' – change to something like: 'that don't follow variations observed in North Atlantic records require...'

Line 109, 119, I recommend avoiding the term 'decline' here, as it may oversimplify the isotopic variations. Consider more precise alternatives like 'decrease' if you want say going to more "negative" isotopic values.

Line 126-127 – 'amplified the influence of zonal Atlantic northern hemispheric drivers of high-latitude isotope relationships' – the use of zonal, atlantic, northern hemisphere and high latitude in this sentence make it unclear what is being said, it should be rephrased to improve clarity.

Line 128 – replace first 'and' with a comma

Line 167, avoid using "exceptional aridity" that can sound subjective or emotive.

Line 168-170 – I am lost here at what sea level change has to do with the climate- perhaps more explicitly link with winter temperatures and summer precipitation

Line 170 – ‘declines in delta18O correspond with rapid sea-level rise associated with meltwater pulse 1b... which led to increased freshwater export from the Arctic Ocean into the North Atlantic known to influence AMOC stability’ – here again you should be more explicit about what you mean. Did the changes in SL and AMOC cause the delta18O decline? And if so how? This was a comment raised in my previous review which has not been improved by the re-write.

Line 181 – I would combine this paragraph with the one above, where no results are mentioned currently

Line 217- ‘are understood to reflect winter precipitation due to groundwater processes’ – understood by who? A reference may be needed here. It is not clear why this interpretation is discounted

Line 219 – ‘which also result in significant isotope depletion’ – I think references are needed here

Line 220 – ‘circulation patterns are well recognized in driving basin-wide patterns in land and ocean temperatures’ – if this is the case, why do the late Holocene changes differ from each other?

Line 241 – ‘weak evidence’ – why is this data weak? if it is not reliable then it should be not used to argue for warm summers, as you do here.

Line 387-389, “a fortuitous aquatic macrofossil...” I would suggest remove the “fortuitous” in case some readers interpret it as “lucky” or “fortunate”.

The reference list is not fully formatted according to submission guidance. Please ensure consistency in reference style (e.g., author names, journal abbreviations, use of italics) and verify that all references are complete and correctly formatted. In addition, there are typographical errors: extra quotation marks at Lines 517, 576, and 583, and a line break issue at Line 533.

Reviewer #3

(Remarks to the Author)

Reviewer #4

(Remarks to the Author)

The authors have provided a revised version of their manuscript which shows that similar decreases in d18O values in two contrasting time intervals (the Younger Dryas and the Late Holocene) can be explained by different climatic mechanisms. In the former case, it is synchronization of Alaska with the North Atlantic Ocean, and in the latter attribution to changes in Pacific Ocean basin atmospheric circulation.

The conclusions remain plausible, and are important results.

I do feel that the manuscript makes a solid contribution to the potential causes of Holocene precipitation d18O variations, but I don't feel that the manuscript conclusively demonstrated the attribution of the Late Holocene variations, and I was left feeling a little underwhelmed with the final conclusions.

The records themselves are really high quality and the paper could even emphasize more strongly (as the authors did in the response to reviewers) that these new records are among the most securely-dated in the circum-Arctic.

Minor comments:

There are a lot of typos in the manuscript. I didn't try to correct them all.

The stable isotope terminology in several cases is incorrect, i.e., it is not possible to have "depleted isotopes". See Table 2.1 of Sharpe's Stable Isotope Geochemistry book for recommendations on proper terminology. "Absolute depletion of d18Op variations" also doesn't make sense. A sample can be depleted in 18O, or depleted in 16O, but it can't be deplete generally. There are other instances where improvement would help the manuscripts presentation.

Figure S2 is not legible, and several of the other Supp figures are of low to poor quality.

I find the authors explanation that meridional flow forced by atmospheric circulation changes in the Late Holocene could produce the very negative d18O anomalies. But I'm still not sure about the attribution. What forced the change? The authors suggested PDO/ENSO type variations, which are plausible sources of the anomalies, but I didn't see a strong conclusion,

i.e., a link to a proxy record of PDO or ENSO that can convincingly show a correlation.

L35: I wouldn't state that lake carbonate and ice d18O are directly comparable, isotopically speaking. One is an indirect precipitation proxy, the other direct.

L39: delete "drift", an outdated term.

L46-7, i think you mean Supp Fig 2? I have no problems with the authors *interpreting* the calcite d18O as a proxy for the d18O of the lake water, which reflects the d18O of winter precipitation. That is well supported. But whether the calcite is truly at equilibrium and has remained so "faithfully" is not so clear.

L48: How large is the region?

L241: What does "weak evidence in glacial moraines" mean?

L246: do you mean "Attribution of such shifts"?

L251: This sentence should be rewritten to something like "Our data show that similar decreases in lake sediment d18O during the YD and Late Holocene can be forced by different mechanisms." As written it doesn't make any sense, because fractionation is not what is being investigated here.

Version 2:

Reviewer comments:

Reviewer #4

(Remarks to the Author)

The authors have done a nice job of improving the paper!

I am satisfied with the revisions and feel the manuscript is ready for publication, after one small change:

The references to d18Op when referring to the lake sediment records is incorrect. They should be d18Oc, which certainly tracks past changes in d18Op, but with a time-varying temperature-dependent fractionation offset and whatever lake-specific isotopic effects are happening. It is ok to use d18Op when talking about inferences about changes in the d18O of past precipitation, but not when referring to the sediments.

Below, we have copied all *comments from the reviewers (black, italics)* followed by **our responses (blue text)**. We provide tracked changes in the revised manuscript accordingly and provide revised line numbers (RL).

Reviewer #1:

In this manuscript, the authors present high-resolution oxygen isotope records from several lakes in Alaska to investigate climate oscillations during the last deglaciation and compare with the late Holocene. They show that the YD mostly represents winter cooling and is closely tied to Atlantic Ocean influences, supporting the existence of strong connections between North Pacific and North Atlantic during the last deglaciation. On the other hand, rapid climate oscillations during the late Holocene were mostly triggered by Pacific Ocean influences, showing no evidence of influence from North Atlantic. They also discuss the implications of the study for recent and ongoing climate changes in North Pacific region.

This is an important study addressing a long-standing paleoclimate question, not only about the connection between North Atlantic and North Pacific, but also the variable nature of this connection. The isotope records presented are valuable and high quality. The interpretations are balanced and robust, supported by the data and arguments. The study design of using lakes with contrasting hydrology and the interpretations of organic $\delta^{13}\text{C}$ records for groundwater flux and moisture conditions are creative and novel. The manuscript is well organized.

Response: We sincerely thank Reviewer #1 for their insightful evaluation. We deeply appreciate your assessment of this study's importance, including recognition of our data quality and the contribution of our creative and original study design. Thank you for guiding improvements of the revised manuscript.

My only major comment is about presentations of dating controls. It is well known that lacustrine sediment records are very difficult to data from lakes in Alaska and Arctic in general, as suitable terrestrial plant materials are hard to find. The authors should be commended for such a large set of radiocarbon dates from these study lakes as presented in SI Table 1. That being said, dating and chronology are critical for the interpretations and conclusions presented in this study, so I suggest that the authors show explicitly the dating control points on the records shown in figures (comment below as well) and discuss any uncertainties accordingly. That way, readers would have a visual appreciation of the robustness of chronology.

Response: We deeply appreciate the recognitions of the extensive collection of dating control points. We agree that dating and chronology are critical and that explicitly showing the age control points in figures provides a visual appreciation of the chronological quality. We have added age points to Figures 1 and 4. Furthermore, we have added a new Supplementary Fig. 3, which shows stratigraphic data on depths scales, including age points so that readers can evaluate age control relative to isotope variations and sampling interval, and lithology (RL 53, 366, 378, 383, 391, 435). We have added text to Methods titled 'Age data and age-depth models' that discusses the interpretations and uncertainties and also references the age-depth models shown in Supplementary Fig 5. (RL 360-415)

Specific comments:

Abstract: mention lake records to make the abstract more informative

Response: Thank you. We substantially revised the abstract and now lake sediment oxygen isotope records is mentioned in the first sentence. (RL 20)

L64: change “guided” to “modulated”?

Thank you, we took this good suggestion (RL 62)

L82-83: Refer Fig. 2b for Neklason Lake record, and d18O from April Fools Lake is not in Fig. 2, but in Fig. 4. Also, should refer Fig. 2a after mentioning Finger Lake here.

Thank you for your detailed review. We have revised the figure references throughout to include lettered panels.

L89: age models: need cite Suppl Figures?

Thank you for identifying this need. We have added specific citations throughout the text to guide readers to the relevant figures in the substantially revised and expanded Supplementary material.

L93: Change superscript 15 to “ref. 15” Thank your detailed review of the figure captions. We have deleted ‘from ref.’ and moved the superscript citation to the record name. (RL 155-163).

L96: “from ~4 to 8 C”. Negative signs have been taken care of by “decline” in the sentence, so not needed here.

Thank you, corrected.

L95-97: unclear. Temperature decline is inferred/estimated from d18O change based on calcite-water fractionation, but what is the role of SS temperature variation here. The former was used for temperature inference/estimate, while the latter is for interpretations. Do I miss anything here? Clarity.

Thank you for pointing out the need to clarify this sentence. We revised to clarify that SST variations were considered in our estimates because of their influence on the $\delta^{18}\text{O}$ value of moisture source vapor. (RL 132-134).

L98: perhaps compare d18O values of winter precipitation/groundwater at the sites and lake-water d18O values calculated from calcite d18O measurements using calcite-water d18O fractionation factor

Thank you for suggestion. In response to this comment and concerns raised by Reviewer #2 we have substantially revised the text and supplementary material to highlight the comparison between measured lake water and calculated lake water values based on equilibrium calcite fractionation factor. The results are shown in Supplementary Table 4 and Supplementary Fig. 2, which are now referred to in the main text (RL 46 and 352)

L101: change to “...sensitivity; however,” This sentence was substantially revised (RL 38-40)

L111-115: add dating control points near curves on figures to support/refute the YD timing discussion here

Done! See detailed response above for additions to Figures 2 and 4.

L115-116: good point! The fact of calcite precipitation suggests that summer water temperature is high enough.

Thank you for noticing this agreement! (RL 139-141)

L121: “*meteoric precipitation*” to avoid possible confusion with calcite precipitation?

Yes, easily misunderstood, good point. However, this section was substantially revised and this sentence removed (RL 175-176).

L134-143: *this paragraph needs to refer to Fig. 3, perhaps different panels according to various descriptions, including to April Fools Lake (Fig. 3b) and synthesis records (Fig. 3c). That would greatly increase clarity and readability.*

Thank you for this helpful suggestion, which we have addressed (RL184, 188)

L155-157: refer Fig. 3 for MWP mention

Thank you, done (RL 173).

L171-172: *Be mindful here. Finger and especially Neklason records have different sampling resolution, so potentially artifact for variability/magnitude. But indeed, April Fools record is a strong proof that variability/magnitude shift around 2 ka.*

Thank you for this very helpful observation that helped guide us to thoroughly reevaluate how to improve the presentation of all our lake $\delta^{18}\text{O}$ records. First, we recognized the value of focusing the presentation of our precipitation $\delta^{18}\text{O}$ records from April Fools and Neklason with Greenland in Figure 2 and removed Finger Lake. Second, we recognized the value in merging the high-resolution sections of the two records into a composite $\delta^{18}\text{O}$ stratigraphy that is now presented in Fig. 1 and 4 and discussed in the Methods and shown in Supplementary Fig. 7. Lastly, we re-conceived Figure 4 to emphasize the comparison between Finger Lake and the April Fools-Neklason composite, including statistical comparison by box plots for time periods to objectively compare means and extremes. Furthermore, sampling intervals for the isotope records are now provided in Supplementary Fig. 3.

L180-183: *very good statements/examples about implications of this study. Well done.*

We sincerely appreciate your recognition of these examples to explain implications. (RL 221-227)

Figure 2: *add dating control points on the figure, perhaps on the top with different colors (such as downward triangles) for Finger and Neklason records, even for composite records.*

Done! We used colored circles that match the color of the records in Fig. 2 and 4.

Fig. 2c: *vertical axis label: change to superscript in m^2*

Yes, on the insolation panel – thank you for your good eye!

L235: *change to “Greenland ice-core”*

Thank you! A hyphen was added between ‘ice’ and ‘core’ when used as a compound adjective throughout the manuscript.

L244: *Fig. 3 caption: indicate the number of organic $d13\text{C}$ records ($n=$)?*

Thank you, n=5 added. (RL 207)

L251: a stray bracket “)”

This figure was substantially revised, and this panel was removed.

L258-259: Livingstone corer can hardly recover sediment-water interface. Either you use a plastic piston corer, or delete the mention as it is not essential for the study.

Yes, thank you for catching this oversight. We revised the sentence to clarify how we collected high-quality surface sediments. (RL 345-346)

L260: we use “endogenic” to describe carbonate, not $d_{18}O$ carbonate, right? rephrase.

Thank you for pointing out this distinction. We made substantial revisions to the sections about carbonate in response to Reviewer #2 throughout the manuscript using the term ‘authigenic’ and this sentence was removed.

L290: an extra “”

Thank you for your keen eye! We removed stray quotes throughout references.

Supplementary Information:

L52, 59, 67: Bayesian

Yes, capitalized.

SI Fig. 3b. vertical axis label “ $d_{13}C$ ”

Yes, for Arolik, thank you, corrected.

SI Fig. 4: n=6 for $d_{13}C$ records need to mention in the main text/figure

Yes, but n=5, Arolik and Dune not used (low resolution) and added a note to caption indicating that.

SI Figs. 5 and 6 are apparently not mentioned in the text, It would be good to do so, when describing the nature/source of organic matter for $d_{13}C$ interpretations.

Yes, thank you for this suggestion that helped guide our revisions in response to concerns by Reviewer #2’s about source characterization. We have substantially revised the Methods and Supplementary material with specific figure citations placed in the main text.

SI Table 1:

3rd column heading: ^{14}C Date (not age here, but “Calibrated Age”)

4th column heading: “...Date (^{14}C yr BP)”, not “years”

Thank you for catching these oversights!

L116: “informal name”. Hopefully next time when the lake was cored again on July 4th, it wont be named “Independence Lake”:)

We appreciate your good humor!

L120: “at two locations”

Thank you, yes, we changed ‘sites’ to ‘locations’.

L292: why don't make it a formal SI Table and formatted as such? The table heading needs proof read, as they are incomplete/inaccurate: such as “Horizons dated”, basal age need “Cal yr BP”? Sampling resolution: “cm” can be moved to heading (should cm be converted to years, as you have age model already?)

Thank you for these very helpful suggestions – all corrections made. We made this Supplementary Table 6 (RL 367).

Reviewer #2:

This paper presents new isotope results for lakes in Alaska that span the Younger Dryas to Late Holocene, providing interesting high-resolution records that are presented nicely in the figures and may eventually advance our knowledge about the Pacific-Atlantic/Arctic teleconnections. As detailed below, while the topic and dataset are promising, the manuscript requires a substantial revision and reorganization to clarify its framing, improve methodological transparency, and provide a more balanced interpretation of the results.

Response: We are deeply grateful to Reviewer #2 for their thorough evaluation of our manuscript and their recognition of the value of our high-resolution records and figure presentations. We sincerely appreciate the feedback about the promise of our manuscript and have implemented substantial revisions to reach its potential that are outlined below. We have carefully reconsidered organization, framing, methods, and interpretation following the reviewer’s suggestions that has elevated the manuscript. We believe the revisions will advance knowledge about winter teleconnection to the North Pacific-Arctic and Northern Hemisphere atmospheric circulation variations. We sincerely thank the reviewer for guiding these improvements.

1. General Comments

1) Chronology and lake reservoir corrections

The age-depth model is central to the paper’s conclusions. However, it lacks sufficient justification and statistical rigor during chronology establishments. I do agree with the idea using the tephra layers with absolute dates to correct the lake reservoir effects. However, I also believe that more details are needed for better clarity:

Response: Thank you for the reviewer’s emphasis on the chronological establishments, which we agree are central to this study. We have a new Methods section for the main manuscript and significantly expanded the Supplementary material to provide additional details and justification that are now cited throughout the main text and described in detail below. The revised text expands on how tephra was used, and the lake reservoir correction (which applied to one of the three lakes), as discussed further below.

a. How the tephrochronology was established for those cores? The authors claim that the tephra correlation is established “Based on similarities” (Line 217 in SI). However, it’s not very clear to the readers what does “similarities” represent. Have you run the geochemical analysis on tephra shards? Which is the routine way for tephra layer correlations. I would suggest the

authors provide more robust information for tephrochronology correlation, either other evidence to clearly show the “similarity” or geochemical analyses results on tephra shards. This will directly influence the reliability of lake reservoir effects estimation and following age-depth modelling for lake April Fools.

We sincerely appreciate the reviewer for these excellent questions that guided our revisions to clarify our tephra identification and correlations. Detailed text on tephra identification and justification, which is fairly well known for this region, is provided in Methods. The stratigraphic positions of tephra in each core are now shown in new Supplementary Fig. 3 and the correlations among the cores are shown in Supplementary Fig. 4. We sincerely thank the reviewer for prompting these additions which has improved the paper through inclusion of detailed stratigraphic and chronological evidence. (RL 371-383)

b. Reasons for the rejection of some radiocarbon dates should be given or detailed. This study presents a massive amount of radiocarbon dating on those cores (which is good) but they also rejected some of those radiocarbon dates but give very little discussion why they did this. Taking core April Fools A15 as an example, the authors rejected more than half of the radiocarbon dates (including the lake reservoir corrected aquatic herb-based dates), but the reason were not given/detailed.

We sincerely appreciate the reviewer’s recognition of the detailed dating that we conducted, which along with the high-resolution data distinguishes our records from existing data in the region. We acknowledge that additional discussion is needed and helpful. We now have detailed discussion in Methods sub-section “Age data and age-depth models” that includes explanation of age rejection of a few ages from the Bayesian age-depth models. This includes clarification that the reservoir corrected aquatic samples in April Fools core A15 were not rejected as ‘age points’ but rather explains why they were not used in the Bayesian age-depth models. Their inclusion did not improve the uncertainty of the model based on terrestrial ages and tephra but verify its accuracy. Because of uncertainty in the temporal variability of the reservoir effect, we take a cautious approach to ensure robust results and limit their use as cross-check for the age-depth model. The ages are now shown in Supplementary Fig. 5a with explanation in the caption. This discussion also explicitly discusses the higher uncertainty in the lower sections of Finger Lake and justification for applying the minimum modeled ages within the 95% uncertainty range now shown in Fig. 4. In response to Reviewer #1’s suggestion, ‘age points’ are now shown in Fig. 2 and 4 and in new Supplementary Fig 7. Furthermore, the stratigraphic position of every age is now shown in Supplementary Figs. 3 in addition to the table of radiocarbon data, Supplementary Table 7. Thank you for your guidance in improving our data presentation and clarity of the chronologies. (RL 384-395)

2) On the interpretations of stable oxygen isotope ($\delta^{18}O$) of the carbonates.

a. Why is your proposed driver the most likely? Lake carbonates $\delta^{18}O$ signals are influenced by multiple factors, including temperature, evaporation, seasonality of temperature/precipitation, and hydrological conditions. While the authors try to classify these three lakes into two groups, namely Precipitation-type and Moisture-type (Lines 53-62), to simplify the interpretation of $\delta^{18}O$ signals, the manuscript does not convincingly justify why those particular factors are the dominant control as claimed (even without reference supporting their classification)?

We thank the reviewer for this constructive feedback to improve these justifications. We have expanded the main text, methods and supplementary material to explicitly evaluate drivers for different lakes and time periods. The revisions also focus on the hydrogeologic mechanisms that emphasize winter climate controls on our groundwater lake water systems primarily recharged by mountain snowpack. We follow reviewer #4's suggestion to use open and closed hydrology terms and refer to our comprehensive regional isotope hydrology framework in Supplementary Fig. 1. In addition to substantial revisions of the introduction and a new "Conceptual Framework", we expanded the Methods in a new subsection "Regional climate and precipitation isotopes controls". (RL 34-113, 282-311).

b. Problem with mixing carbonate types. The manuscript does not clarify the type of carbonates being analysed. From the method section, it seems that the fine fraction (<32 µm) was used for isotopic works. However, this fine fraction seems to be a mixture to me, and this carbonate source ambiguity raises concerns about the geochemical integrity of the proxy signal. For example, what is the proportion of biogenic, inorganic, or even detrital carbonate in the finest fraction (<32 µm)? Are there sedimentary or geochemical indicators that confirm the origin of the carbonates?

Thank you for pointing out the need to better clarify our methods and validate the geochemical integrity of the proxy signal. We acknowledge the importance of this concept and have rephrased several sentences in the introduction to clarify our analysis of bulk inorganic carbonate of authigenic origin (i.e., formed where they found – in this instance, each lake). We expanded the Methods with a new section "Lake sediment lithology and carbonate isotopes" and the Supplementary material to show sedimentary and geochemical evidence for no significant detrital contribution or changes in sedimentary processes that would affect the purity or isotopic composition of bio-induced bulk carbonates. In addition to assessing carbonate grain appearance by microscopic observations they include 1) pH, alkalinity, biological productivity, and lake water temperatures that support carbonate precipitation caused by $p\text{CO}_2$ during photosynthesis, summarized in new Supplementary Table 1, 2) verification by isotopic equilibrium experiments described in Methods and shown in new Supplementary Table 4, 3) stratigraphic profiles that show non-authigenic sedimentary components comprise a minor fraction of total sediment composition, in new Supplementary Fig. 3, and 4) no evidence for detrital carbonate sources or mechanism of delivery to our lakes. We thank the reviewer for guiding these important improvements to the manuscript (RL 39, 50-52, 312-359)

c. There is currently no discussion of possible isotopic offsets between different carbonate fractions. This is a crucial omission, as differences between, for example, biologically precipitated carbonate and inorganic calcite can introduce systematic biases. Is there any isotopic offset between different grain-sized fractions, between fractions <32 µm and 32-63 µm? Change in the carbonate source can influence the isotopic signals of the carbonates, which is independent of climate change.

We appreciate the constructive suggestion and address it fully in the revision. We have expanded Methods and Supplementary material to explicitly address our precautions to avoid mixing authigenic bulk carbonates of inorganic or bio-induced origin from biogenic carbonates (e.g., gastropods, ostracodes, Psidium). The revisions clarify that large particles of biogenic carbonates were separated by sieving and that we analyzed the finest fraction to minimize their potential effects. The finest fraction is then shown to be in isotopic equilibrium with lake water (as

discussed in the previous response), and we clarify that the need to mix small size fractions in order to achieve adequate samples was required for <1% of all samples measured. Further, we conducted a series of bulk carbonate size fraction tests and in no instance is a significant deviation of isotopic values for fine grained size fractions observed, now shown in Supplementary Table 3. We are grateful to the reviewer's comments that have guided our method clarifications which strengthen our climatic interpretation of our lake carbonate isotope signals. (RL 345-359)

I would recommend:

i. Consider running modern calibration studies on modern water or carbonates samples to enhance the interpretation of the isotopic signals of carbonates (e.g., comparing $\delta^{18}\text{O}$ of precipitation, lake water, and carbonates with climate parameters).

We agree with this excellent suggestion and appreciate the opportunity to clarify that these calibration studies had been conducted. We acknowledge their presentation required improvements as described in our response to the previous comments above. We thank the reviewer for guiding these improvements.

ii. If only bulk carbonates were analysed, the potential impact of this choice on the paleoclimate interpretation should be clearly discussed and acknowledged as a limitation.

We appreciate this concern. As discussed above, the main factors that can adversely affect climate signals recorded by bulk carbonates have been evaluated and determined to be of minor impact in our systems. Following the reviewer's suggestion, we take the opportunity to explicitly state that in this hydrogeologic setting, authigenic bulk fine-grained carbonate $\delta^{18}\text{O}$ from groundwater lakes provides exceptional continuous representation of lake water $\delta^{18}\text{O}$ that can be well dated and thereby provide climatic records of unprecedented temporal resolution. (RL 40-48)

3) On the interpretations of bulk organic stable carbon isotope ($\delta^{13}\text{C}_{\text{org}}$)

The authors interpret the $\delta^{13}\text{C}$ values of bulk organic matter primarily as a proxy for precipitation variability. However, this interpretation appears overstated and does not adequately consider alternative influences. The C/N ratios reported (generally <15) suggest that the primary source of lake organic matter is aquatic algae. This implies that $\delta^{13}\text{C}$ variations likely reflect lake-internal productivity processes rather than external hydrological inputs, at least not only external processes. Aquatic algae assimilate dissolved CO_2 , and the isotopic signature of that carbon pool is influenced by lake water $p\text{CO}_2$, which itself is affected by temperature, biological activity, and CO_2 exchange with the atmosphere. These internal carbon cycle dynamics can significantly modulate the $\delta^{13}\text{C}$ signal, making it problematic to directly attribute isotopic shifts to precipitation changes alone.

Thank you for pointing out the need to better explain our consideration of all alternatives in more detail. According to the reviewer's suggestion we have expanded Methods with a section titled "Lake sediment carbon and nitrogen organic matter" and revised Supplementary material and Fig. 3 to show that during the period of $\delta^{13}\text{C}$ decline and lake level rise, for which we have determined using C:N that the organic material is of aquatic origin, the resulting interpretation boils down to productivity or hydrology. Supplementary Fig. 9 now shows a measure of productivity (e.g., organic content as determined by % LOI-550°C) in addition to C/N ratios,

with either no trend or an increasing trend, which would result in either no change in $\delta^{13}\text{C}$, or change in the opposite direction (increase) than the data show. The revisions now explain how we ruled out this alternative explanation for this time period when all lakes respond with $\delta^{13}\text{C}$ declines. However, subsequent Holocene changes diverged among the lakes, likely due to these internal productivity processes, which is acknowledged. In summary, our data show that during this specific time of major hydrologic change, external hydrologic processes were more significant than internal carbon source or productivity processes. (RL 177-196, 416-447)

In its current form, the manuscript risks overinterpreting $\delta^{13}\text{C}$ as a direct climate signal, and I recommend the authors reconsider this conclusion and broaden their interpretive framework. We deeply appreciate the reviewer's thoughtful comments that have helped to significantly improve our explanations of the full scope of alternatives strengthening support for our arguments in favor of the proposed interpretation (see discussion in above comment and associated revisions to text and supplementary material).

4) *Mismatch between the paper's stated goals and its actual contributions.*

While the manuscript presents interesting datasets from multiple lake records, I find that it ultimately does not achieve the stated goals outlined in the introduction or title. The authors claim to resolve key questions about climate connections between the Pacific and Arctic/Atlantic, yet the evidence presented—particularly regarding the $\delta^{13}\text{C}$ and $\delta^{18}\text{O}$ interpretations—remains ambiguous and open to alternative explanations that are not adequately explored. This ambiguity in the climatic significance of the proxy signals, combined with uneven methodological transparency (e.g., tephrochronology, lake reservoir corrections, unseparated carbonate fractions), undermines the strength of the conclusions. As a result, the study does not convincingly advance our understanding of teleconnections between Pacific and Arctic/Atlantic in the way it aspires to.

Thank you for pointing out the need for clearer methodological transparency, and greater evaluations of alternative explanations. According to the reviewer's suggestion, the revisions have clarified our chronological methods and the basis for our interpretations of carbonate source and isotopic signal. Expanded discussion of alternative explanations of our evidence strengthens confidence of the hydroclimatic interpretations based on robust independent age chronologies, allowing confident comparisons between North Pacific variations with those in the North Atlantic based on the Greenland Ice core. While our work builds on previous studies, it significantly improves chronological control, record continuity, and temporal resolution based on the robust bulk sediment carbonate isotope proxy. We have expanded the text to include a "Contextual Framework" where we specifically explain our hypotheses that we believe advance understanding of seasonal atmospheric teleconnections and their relationship to $\delta^{18}\text{O}$ signals through time. (RL 95-113)

More supporting evidence is needed to back up that the lakes reflect winter conditions (references are not given in several places), and the authors often argue that changes reflect temperature, without explaining why precipitation source and atmospheric circulation is discounted. Furthermore, the lake records show very different variations during the late Holocene, but rather than acknowledge this and explain why this might be (and give a

reasonable explanation of why they still have a climate signal) the authors discuss the high variability as being significant.

We sincerely appreciate this insightful review and have accordingly substantially revised the organization and data presentation to support our conclusions regarding the role of North Atlantic forced temperature during the deglacial in contrast to atmospheric circulation during the late Holocene. As described above, we have expanded the Methods and Supplementary material to clarify and substantiate that our groundwater lakes have sensitivity to winter precipitation $\delta^{18}\text{O}$ and revised Fig. 1 to focus on the Greenland comparison with our composite records of April Fools Lake and Neklason Lakes, now described in Methods and illustrated in Supplementary Fig. 7. We now present a conceptual framework that discusses how hypotheses regarding temperature and atmospheric circulation changes can be tested using $\delta^{18}\text{O}$ records and reorganized other sub-sections accordingly. We revised Fig. 4 to focus on the comparison between our record of winter precipitation $\delta^{18}\text{O}$ and summer evaporation from Finger lake with additional statistical analyses by box plots to emphasize the comparison between means and extremes for the YD and late Holocene. Reviewer #4 raised similar concerns that we address further in detail below.

Finally, at some points in the paper the reader is left behind because the links between ideas are not explained fully. In parts the text requires reorganisation and rewriting, as comparisons with other records or supporting evidence is spread over several paragraphs.

Thank you for your valuable feedback about the need to carefully evaluate and revise the links between ideas throughout the paper. We substantially reorganized and rewrote the main text and expanded the Methods to improve the logical flow of ideas and make the concepts and explanations easier to follow.

2. Moderate comments

Main section:

Line 53-62 – there is just one reference provided for the interpretations in this paragraph, do the interpretations all come from here? Are there modern observations or data to support this? References or data are needed particularly to support the interpretation of these being open/closed basins and the water residence times, which are the basis for the interpretation about seasonality.

Thank you for the opportunity to explain that this reference by Kikuchi (2013) is an exhaustive U.S. Geological Survey hydrogeologic investigation of our study region that provides modern climate, hydrology, and isotope observations, data, geologic framework and hydrologic modeling in support of our hydrologic interpretation of the study lakes (in addition to our data). The report shows strong linkages between lake water isotopes and groundwater with related hydrologic setting and climate. The information in this report is specific to our study area as is the work by Bailey et al. (2019) who analyzed weekly $\delta^{18}\text{O}$ of precipitation 2005-2018. The major conclusions from these studies are summarized in the main text and expanded Methods to the extent allowed by length limitations. (RL 49-60, 261-281).

Line 67-68 – ‘Contemporary 18O precipitation reflects seasonal air temperatures and shifting moisture sources in addition to topographic rain-out effects’. This sentence sums up some of my concerns later in the paper where changes in the isotopic records are interpreted in certain ways, such as temperature, without considering the other possible causes given here.

We acknowledge the need to improve our contextual framework for the unique approach we take with respect to multiple controls on precipitation $\delta^{18}\text{O}$, which we have addressed throughout the revised text. By using the combined lake datasets by comparison to North Atlantic records, we identify changes in winter atmospheric teleconnections that best resolve $\delta^{18}\text{O}$ signals through time by different controls. For this region, as this conceptual framework explains, modern controls are primary moisture source and atmospheric circulation by teleconnection related to North Pacific ocean atmospheric dynamics, as demonstrated by several published studies we have referenced. Such signals are not recorded in ice-core records from Greenland. However, this mechanism is a highly unlikely for the YD, because we show correlative signals in both regions during this period. This leads to our conclusion that the most parsimonious climate driver involves and teleconnection resulting from North Atlantic ocean dynamics by causing colder hemispheric high-latitude atmospheric temperatures. Ultimately, we argue that similar signals in precipitation isotopes can result from condensation fractionation effects through either high-latitude cooling or long-distance meridional vapor transport, and that these processes can be distinguished by similarities and differences in records from different regions and hydrologic sensitivity. This explanation is now reflected in the revised text. (RL 95-11, 116-152, 213-228, 239-257).

Line 77 – ‘resolving past seasonal changes’. In the next section you explain why the lakes reflect winter changes but that is not explained well here. I think add to the end of this section more detail and references supporting how you know it reflects winter season changes.

As described above, we expanded the text, Methods and Supplementary material to explain why the lakes reflect winter changes and this sentence was rewritten.

Line 81-106 – These two paragraphs could be reorganised – it is quite confusing currently having comparisons with existing records spread over two paragraphs, and the statement about the YD being a winter phenomenon here only makes sense with the evidence shown in the following paragraph

Yes, thank you, these two paragraphs have been rewritten as described above we have reorganized and expanded the main text to improve the logical flow of ideas and better link concepts.

Line 98-100 – ‘Our YD delta 18O carbonate records largely reflect winter season cooling’. How do you know it reflects cooling, could winter season circulation changes also have had an effect?

Thank you for your insightful question. We revised the text to include these alternatives and suggest that both can be true if changes in winter circulation led to colder conditions. (RL 99-103)

Line 102– ‘Our new data substantiate that the YD in Alaska was most sensitive to wintertime cooling,’ The phrasing of this could be improved as currently the YD is sensitive, rather than the YD being a time period and the climate or winter season climate being sensitive. The references given later for the YD summer being warm should also be put here (currently down at line 112) as should the evidence from the lakes themselves (high productivity and marl accumulation) currently at line 115. This would strengthen the argument that the record is reflecting the winter rather than summer. Consider also my point for line 98 about the interpretation of this as cooling rather than atmospheric circulation, how do you know this?

Thank you for your valuable feedback about correctly phrasing modifiers. We re-wrote the section on YD, including the text and references about summer conditions that strengthen the arguments that our records record winter precipitation $\delta^{18}\text{O}$. (RL 117-152)

Line 126 -129 – references are needed here to support this interpretation

Thank you, references added to Methods to support our interpretation (RL 463-496)

Line 143 - I am not sure if moisture shifts is the right term here for what you describe – a shift would suggest to me a change in the distribution of moisture across the area (so maybe some areas having more and others less). But as they are all changing together, are they not just experiencing enhanced moisture?

Thank you for your thoughtful consideration of accurate terminology. We agree and re-wrote this section using specific terms such as precipitation, surface runoff, groundwater flux etc. (RL 177-182)

Line 146 -148 – ‘both lake types indicate the warmest year-round temperatures and possibly the greatest aridity of the records’. In the Main section above when describing the causes of oxygen isotope changes the atmospheric circulation paths are described – how do you know therefore that the oxygen isotope changes are due to temperature changes and not changes in atmospheric circulation? Both types of lake are influenced by the precipitation, so this seems more likely to be causing shared changes?

We sincerely appreciate your thoughtful insight. We agree that both lakes share common changes in the $\delta^{18}\text{O}$ of precipitation. We re-wrote this section to clarify that at Finger Lake positive extrema due to evaporation during the summer in response to warm temperatures. For Nekalson, positive extreme could be shift in rain/snow balance to rain, also in response to warmer winter temperatures. (RL 166-168)

Line 148 – ‘However, with winter experiencing the most significant warming, high delta 18O...’ Is this based on the cold YD temperatures? How do you know this?

Yes. Based on the winter sensitivity of the groundwater lake $\delta^{18}\text{O}$ proxy, we interpret the rapid increase during the YD to reflect increased winter temperature although we now consider alternative interpretations such as a higher ratio of rain to snow or increased moisture sources from the Bering Sea related to melting sea ice, which are both consistent with winter warming. (RL 166-168)

Line 146 -159 – In this paragraph I feel like a few steps are missed in the explanation – I think you explicitly need to mention that it is thought that freshwater export may have weakened the AMOC (if this is what the references here say) and say how that would have caused warming/zonal circulation in Alaska. And was it the Bering Strait flooding or AMOC changes that you think caused the changes in Alaska? If you don’t know say more clearly that there are a few potential explanations, or different causes at different times (as suggested by figure 3).

Thank you for this suggestion which helped our conceptual framework and revisions of Fig. 3 for this complex period of change. We rewrote this text to link the steps within the explanation of how weakened AMOC and opening of the Bering Strait led to warming in Alaska. (RL 172-176, 194).

Line 163-164 – ‘This contrasts with Greenland, which experienced continuous temperature sensitivity to AMOC’. A reference is needed for this.

Thank you, reference added.

Paragraph starting 174– here you change the interpretation from oxygen isotope variations being caused by temperature (during the deglacial) to atmospheric circulation. This interpretation is given very certainly, but without many references or explanation of why this change would occur. Furthermore, why are the records all showing different patterns if they reflect atmospheric circulation that would be the same across the area?

Thank you for your suggestion to better explain our interpretations. The revision now clarifies how changes in temperature and atmospheric circulation affect oxygen isotopes of precipitation in the contextual framework and considers alternative explanations, which has strengthened our interpretive arguments. Our focus in this paper is on broad late Holocene trends in comparison to the deglacial. Because of the complexities in regional expression in response to atmospheric circulation shifts in this topographically varied region, detailed examination of late Holocene variation is beyond the scope of this manuscript. We hope this will be the topic of future papers.

Line 180– ‘For example, our records document greater extremes in pre-historic baselines that help evaluate recent marine heat waves and associated sea bird, and fish die-offs.’ This is a rather general statement – do you mean greater extremes than the 20th century variations? Greater using which variable? If you are able to say this then it should be supported by an example of how your results can be used to evaluate these impacts, or at least how modern observations of oxygen isotope values compare with the late Holocene range.

Thank you for these suggestions and we appreciate the suggestion to be more specific. We have revised the text to explain that we mean greater extremes in $\delta^{18}\text{O}$ as a proxy for ENSO-PDO and how modern variability compares with that during the Holocene. We also identify the 1976 PDO shift in Fig. 2 and 4. (RL 85-88)

Line 197 -199 – the links with precession and global changes should probably be introduced before the conclusion section. Without more explanation I don’t understand what is meant by this sentence, or how you have come to this conclusion. This is also discussing the late Holocene changes, which as I mention above are not consistent between your records, which I think is problematic for then making conclusions about the late Holocene climate.

Thank you for pointing out the need for more explanation. The links between ENSO and insolation are now explained in the conceptual framework with links to the concept of latitudinal gradients (RL 106-108, 247)

Line 206 -207 – ‘records show Late Holocene climate shifts comparable to those of the YD’ – you interpret that the isotope changes during the YD are caused by temperature, and during the late Holocene by circulation changes. So I am not sure you can then say they are comparable – it implies YD-scale temperature fluctuations were occurring in the late Holocene.

We rewrote these statements to clarify that the magnitude of $\delta^{18}\text{O}$ changes are comparable, and not temperature. (RL 24, 234)

3. Minor Comments

Line 3, the title is concise and readable, but it does not fully capture the scope or key findings of the study.

Thank you and we agree. The title has been substantially modified to better reflect the study scope and key findings

Line 39 – ‘inform abrupt climate change’ – the word inform doesn’t fit here, perhaps ‘inform our understanding of abrupt...’ would work better

Thank you, we rewrote this sentence. (RL 257)

Line 52 – ‘unresolved timing’, this could be better phrased as ‘uncertain chronologies’.

Unresolved timing requires more detail about what the timing is related to.

We rewrote this sentence. (RL 38-40)

Line 66 – ‘oriented by coastal topographic barriers that were continental ice sheets during past glacial periods’, this phrasing is confusing. Do you mean the topographic barriers were shaped by the ice sheets? Or the barriers are currently topographic but there were ice sheet barriers in the past?

We rewrote this sentence (RL 63-65)

Line 73 and 74 – The use of the word ‘reflect’ here I think is unusual but perhaps ok. I would consider ‘cause’ instead

We rewrote this sentence without using this term (RL 78)

Line 74-76– references needed here

Added ENSO-PDO references (RL 81)

Line 107– add ‘timing and nature’ or something similar, as most of the discussion here is about the character of the YD rather than the timing.

Agreed (RL 34)

Line 123 – ‘with a strong consensus for there having been an Early’

We rewrote this sentence (RL175-176)

Line 210 -211 – reference needed

We rewrote this section and removed this sentence.

SI Lines 292-293, Lake water pH or conductivity are needed. Lake water pH or conductivity measurements should be presented as geographical setting or background information. It can be useful for the interpretation of the carbonates $\delta^{18}O$ signals as disequilibrium offset increases at high alkalinity.

Thank you. This information is now shown in Supplementary Table 1.

Reviewer #3

We sincerely thank Reviewer #3 for their time and efforts contribution to this review.

Reviewer #4

This is a nice paper that presents three new detailed carbonate $\delta^{18}\text{O}$ lake sediment records from Alaska. They records are of high quality and important for understanding regional paleoclimatic variations, and their potential links to abrupt climate changes in the North Atlantic ocean over the deglacial period.

Response: We deeply appreciate Reviewer #4's positive assessment of our paper and recognition of its contribution to determining linkages to abrupt climate changes in the North Atlantic.

The major comment I have for this paper is that, while great records and important, I don't see what new discoveries were made about climate dynamics in the North Pacific. Some of the statements about the variability in the Holocene (after the more clear B/A and YD intervals) were not fully reasoned out, and their differences to the Mt. Logan ice core were not explained. I can certainly see why insolation might be a control, as suggested, but I don't see the reasoning that supports such an interpretation, or why the Late Holocene $\delta^{18}\text{O}$ record should be interpreted differently than earlier periods.

Thank you for pointing out the need to better explain our reasoning about the role of North Pacific climate dynamics and its influence on the isotopic records. We appreciate the opportunity to clarify late Holocene $\delta^{18}\text{O}$ signals in contrast to the deglacial. To clarify this important distinction, we reorganized the introduction and refocused the sub-sections in the results. We also revised the title of the paper and re-conceived the figures to better clarify how our records build upon previous work. Our substantial revisions clarify that the Mount Logan record established the importance of late Holocene moisture source and atmospheric circulation variations. However, the uncertain deglacial chronology beyond AD 800 (e.g., correlated to Greenland by sulfate peak clusters) severely limits any interpretation of the deglacial record. Our data, by providing excellent independent chronological control for multiple lake cores verifies the existence of an atmospheric YD signal in Alaska that we propose is best explained by winter temperature. This is a particularly important discovery in the North Pacific-Arctic (Alaska) since identifying this event and determining its linkage with the Atlantic has previously been difficult to determine, creating uncertainty about the interplay of Atlantic and Pacific teleconnections through time. We have also clarified our reasoning for an insolation control within our new framework paragraph. This explains how summer insolation is a known driver of ENSO/PDO dynamics – this is shown by modeling and extensive number of proxy datasets. Furthermore, shifting seasonal insolation trends during the Late Holocene is also understood as a driving mechanism for the global neoglacial period, as reflected by glacier mass balance in Alaska and our isotopic records. We hope these clarifications and discussion address the reviewer's questions.

In summary, this is a solid regional-level characterization of south central Alaskan Holocene paleoclimate from lake sediment records. The time series are of high quality and will be an important contribution to our understanding of regional paleoclimate. But the manuscript

doesn't yet make a convincing case on the hemispheric or global forcings of the paleoclimatic changes, or why they are significant for our understanding of north Pacific climate dynamics. We sincerely appreciate the reviewer's recognition of the quality and importance of our data and thank the reviewer for helping to guide the improvements in making our case in new and revised text. We have focused the revised manuscript on clarifying how the records show a shift in teleconnections by different hemispheric and global forcing mechanisms through time and the implications for understanding isotope systematics and the role of seasonality and circulation patterns with regards to abrupt climate change. Furthermore, we emphasize that climate variations in this region have significant and broader implications, as changes in the Aleutian Low and ENSO that influence terrestrial and marine ecosystems over much of North America and the entire North Pacific basin.

Minor points

I don't think the authors need to redefine open and closed basin lakes as "precipitation-type" and "moisture-type". Just the description of open and closed and their relative sensitivities to precipitation $\delta^{18}O$ would suffice.

Thank you, we agree and revised to open and closed terms. (RL 52, 56)

I presumed in the introduction that the authors would come back to the Pacific climate modes and show some linkages between proxies for those and the lake sediment records. The lake sediment oxygen isotope records were only qualitatively linked to these to these Pacific Ocean climate modes, without any conclusive evidence of them being important forcings.

Thank you for pointing out the need to explain that these mechanisms have been well documented by numerous previous studies of lake sediment carbonate oxygen isotopes, as well, as Mount Logan and throughout western North America. We have re-written the introduction and included several of these citations. We have added discussion of the April Fools data which provides conclusive evidence for the importance of North Pacific forcing during the most extreme PDO-regime shift (e.g. 1976) of the historic period by recording a coincident large negative excursion now shown in Fig. 2 and reconceived Fig. 4 to focus on the seasonal information provided by our records and broad-scale millennial patterns. Detailed analysis of the complex topic of late Holocene event-scale analysis is beyond our space limitations here and is the subject of future paper in progress. (RL 78-92)

The authors can describe the source of the carbonate in the lakes up front. Are they measuring authigenic calcite, shells, or something else?

We thank the reviewer for emphasizing this need that was also brought up by Reviewer #2 above, where our detailed response of revisions made to the Methods and Supplementary material is provided.

In line 174, the authors did not provide their evidence that late Holocene oxygen isotope values would have a different interpretation than during the deglacial period.

Thank you for pointing out the need for a better organized and clear explanation of the evidence for the different interpretation of late Holocene oxygen isotopes values from those of the deglacial. The revised the introduction, conceptual framework, and methods have been designed

to clearly explain our evidence for different isotopic controls (e.g., winter temperature versus moisture source and atmospheric circulation)

Further, Rayleigh distillation is happening in all cases across all time intervals, and therefore is not unique to the Late Holocene, so further evidence needs to be provided to support the authors reasoning.

Thank you, yes, we agree. We revised our use of the term Rayleigh distillation to include by what mechanisms driving negative isotope excursions, either by temperature of vapor condensation related to colder atmospheric temperature or by increased long-distance northward sub-tropical vapor transport during meridional circulation patterns related to North Pacific ocean modes. (RL 103, 300, 306)

Similarly, the manuscript would benefit (in line 198) a better description as to why orbital precession is now an important control over the Holocene. Is this backed up by model runs? And are the authors referring to local summer, winter, or other season for insolation?

Thank you for guiding our revisions to clarify the roles of precession during the Holocene and its influence on North Pacific modes that we addressed above with respect to insolation. (RL 110, 150).

In figure 2 the authors show the Mount Logan ice core record, which spans the entire time interval covered by the lake sediment records. So, what do the new lake sediment records provide for our understanding of deglaciation paleo climate that was not apparent in the Mount Logan ice core record? There are some obvious differences, with the Mt. Logan $\delta^{18}O$ record having a relatively non-trending Holocene, whereas the lake records show more change. What would explain the discrepancy?

We thank the reviewer for pointing out the need to clarify the strengths and chronological limitations of the Mount Logan ice core record. Beyond AD 800, its chronology is highly uncertain, and across the deglacial interval it is based on similarity of sulfate peak clusters with those in Greenland (Fisher et al., 2008). Thus, our new lakes records substantially build upon the strengths of the Mount Logan record with improved independently dated records that can now confidently correlate deglacial climate shifts to Greenland. The differences in Logan's Holocene trends are related to unique features of the high elevation ice-core that is the subject of several previous publications cited, and while beyond the scope of this paper is a planned subject of future work. Thank you for your comments that guided our improvements of the data presentation and explanations.

Responses to reviewer comments received 3 Sept. 2025, regarding revised manuscript NCOMMS-25-12495A to Nature Communications

Title: Shifting Winter Atmospheric Teleconnections to the North Pacific reconcile Younger-Dryas and Holocene $\delta^{18}\text{O}$ signals

Below, we have copied *all comments from the reviewers (black, italics)* followed by *our responses (blue text)*. We provide tracked changes in the revised manuscript accordingly and provide revised line numbers (RL).

Reviewer 1:

The authors have carefully addressed my concerns on the earlier version of the manuscript and have significantly revised and improved the manuscript. I have no major comments.

We sincerely thank Reviewer #1 for their supportive evaluation and detailed editorial suggestions.

I have some editorial suggestions:

Figure 3: Axis label “Age” should be “Age (cal yr BP)”? Thank you, done.

Ref. 51 and 83 are the same reference. Thank you for catching this error, corrected.

Supplementary file:

Table S1: Upper cases and lower cases are inconsistent. For example, “Temperature range”, “Specific Conductance”, “Dissolved oxygen”, etc., etc. (similar U/L cases issues with other tables as well) All are corrected to U/L format and all tables have been checked for consistency.

Table S5: Change “210Pb Age AD” to “210Pb Year AD” or equivalence (2015.2 is not age (yr old), but date) Thank you, done.

Table S6: “Dated horizons”? Thank you, done.

Fig. S6: Pollen concentration needs unit Thank you, grains/cm²/yr, done.

Reviewer 2:

The manuscript has been improved substantially by the changes made by the authors and it reads better than it did previously. The revision has increased the transparency of the methodology and strengthened the justification for its robustness. The findings related to the YD and Early Holocene are convincing and interesting, and well presented in the figures. The study is strengthened by the additional analyses or reorganization of the manuscript, and the work has the potential to be impactful.

That said, I am remain unconvinced about the significance of the variations for the late Holocene, which is a section of the manuscript that either requires better justification or for the conclusions to be altered.

Another concern is about the organization and narrative flow of the revision. The overall organization of the manuscript is still challenging to follow, especially for readers not familiar with the study area. While the manuscript has been improved, it still requires some sentences or paragraphs to be rephrased for clarity or to provide more explicit explanation of the points that are being made.

We are sincerely grateful to Reviewer #2 for their recognition of the improvements in organization, transparency, justification and for their positive assessment of the findings and

potential impact. We appreciate their feedback and have carefully reconsidered the justification for the late Holocene variations. Following this feedback, we have also improved the overall clarity of the paper, addressed through reorganization of the introduction and including more explicit explanations of points of interpretation. We are very grateful to the reviewer for guiding these important improvements that facilitate understanding the main points of this research by a broad spectrum of readers.

Major point:

1. Although the authors have substantially revised the structure and flow in response to my previous comments, I remain concerned that the manuscript's organization is still not optimal. The introduction does not follow the general structure expected in a Nature Communication manuscript, which should clearly present the significance of the topic and the central knowledge gap. Given the text constraints, there may not be space for an extensive review of prior work, but it remains unclear what the main knowledge gap is and how this study addresses it. I recommend a sharper framing and restructuring of the introduction to highlight both the novelty of the work and its broader significance. For example, in Lines 35–48, the introduction begins by emphasizing the novelty of the research and technical details, which is not an effective way to start.

We appreciate this excellent feedback and agree the framing can be sharpened by following the suggested structure. We have substantially revised and reorganized the introduction to more clearly present the topic in the expected order starting with significance, followed the central knowledge gap, how this study addresses it highlighting the novelty, and broader significance. Overall, the revised introduction begins generally and continues with greater specificity and circles back to broader significance.

2. Following the Introduction, there is a section titled 'Conceptual Framework.' Its purpose is not entirely clear, and the content appears to be more introductory than what is currently presented in the Introduction. I recommend merging the 'Introduction' and 'Conceptual Framework' into a single, stronger Introduction, unless the journal requires them to remain separate.

Thank you for this very helpful observation. We merged the conceptual framework with the introduction as suggested which we believe has significantly strengthened the reorganized introduction. We thank the reviewer for guiding these important changes.

3. In the previous review I questioned why the records differ for the late Holocene section if they are reflecting regional climate and atmospheric circulation, and this has not been explained. I don't think this has been addressed by inclusion of the ranges in figure 4 or the additional conceptual framework (you state at L105 'Within Alaskan records, differences in timing and magnitude of delta18O changes imply variability in both temperature and precipitation resulting from atmospheric circulation patterns' – but across a relatively small region such as this changes in temperature and precipitation would be the same surely?). I cannot find anywhere else this has been discussed. Perhaps a comparison of the records showing just the late Holocene section in the supplementary file would show that they co-vary but have different magnitudes, it is hard to see if this is the case from the figure, which could be explained by local factors. Otherwise I don't see how you can conclude that the lake records are reflecting late

Holocene changes in atmospheric circulation (as you do at line 220) as this would alter climate in a similar way at all the lakes.

We sincerely appreciate the reviewers excellent questions that guided our revisions. We now clarify this issue by noting that $\delta^{18}\text{O}$ variations may differ between nearby open and closed lakes because they reflect different seasons (RL 202-203, 247-250). The revisions better illustrate our objective to utilize differences between the open and closed records to evaluate changes in seasonality through time and explains that the sensitivity of closed and open lakes varies with regard to summer and winter climatic conditions, respectively. In addition, the revised introduction explains the rationale for selecting open and closed lakes as a way to specifically explore independent summer and winter climatic variations that may have differences (RL 66-79). Finally, we revised the Fig. 4 annotations and caption to emphasize the winter and summer seasons.

4. The revisions have introduced valuable new text, data, and figures (primarily in the Supporting Information), which strengthen the manuscript scientifically. At the same time, this restructuring has created some clarity issues that were not present in the original submission. I would encourage the authors to emphasize the central findings more prominently in the main text and streamline the methodological details into the Supporting Information, while ensuring a clear and logical flow. This adjustment would help the manuscript become more concise and accessible to a broad readership, and ultimately enhance its impact and visibility.

Thank you for the recognition that the revised Supplemental Information strengthens the manuscript and for these suggestions that help to further enhance impact. We have revised the main text to more prominently emphasize our central findings. Our revisions of the introduction included streamlining methodological details to better emphasize important strengths of the data (chronology, resolution, continuity, seasonal sensitivity) and to explain why late Holocene variations may differ between open and closed lakes (as described above).

5. Results and discussion structure. The results section currently contains extensive commentary that reads more like discussion (For example, Lines 126-134). Conversely, the discussion section is largely a continuation of further discussion. I would recommend merge results and discussion into a single, well-structured section. Thank you for pointing out this discrepancy and we have changed this heading to “Results and Discussion” as suggested. It is one single section structured into three sub-sections that are logically organized by time-period (deglacial, transition and Holocene). We thank the reviewer for this very helpful suggestion.

Moderate/minor points:

Lines 26-28 – this is not a clear sentence, consider rephrasing

We revised this sentence to better highlight our findings and their significance to “Our new records are among the most reliably dated records yet produced in the circum-Arctic and show that similar decreases in $\delta^{18}\text{O}$ of winter precipitation during the Younger-Dryas and late Holocene were driven by different atmospheric teleconnections” (RL 26-28)

*Lines 52-53 – the c and p on the delta18O could be explained using brackets here
The 'c' is defined on line 44 (RL 62) and the 'p' is defined on line 35 (RL 41).*

Line 64-67 - this is not a clear sentence, consider rephrasing
This has been revised into several sentences to clarify the intended meaning (RL 86-95)

Line 77 – ‘associated with intensified’ – often words like an, or the, are missing through the main text. Please check this here and elsewhere.
Added ‘an’ and checked for similar missing words throughout.

Line 87-89 – references needed for the previous proxy studies
Added references (RL103)

Line 90 – ‘would imply unrealistic higher temperatures’ is it unrealistic in magnitude? or just that you expect the LIA to be colder?
Revised to clarify that we mean warmer temperatures would be inconsistent with the broad knowledge that this is a relatively cold period (RL 104-105)

Line 98, I don’t think “YD is the most recent stadial period”
This phrase has been deleted.

Line 104 – ‘that are not observed in the’ – change to something like: ‘that don't follow variations observed in North Atlantic records require...’
Thank you, we made this suggested change (RL 46-47)

Line 109, 119, I recommend avoiding the term ‘decline’ here, as it may oversimplify the isotopic variations. Consider more precise alternatives like ‘decrease’ if you want say going to more “negative” isotopic values.
As suggested, we have changed this terminology to decrease here, and in other similar instances. (RL 125)

Line 126-127 – ‘amplified the influence of zonal Atlantic northern hemispheric drivers of high-latitude isotope relationships’ – the use of zonal, atlantic, northern hemisphere and high latitude in this sentence make it unclear what is being said, it should be rephrased to improve clarity.
Thank you. We rephrased this sentence and specifically simplified this phrase to “...that shifted the polar jet stream southward.” (RL 133-136)

Line 128 – replace first ‘and’ with a comma
Done (RL 136)

Line 167, avoid using “exceptional aridity” that can sound subjective or emotive.
Changed to unprecedented (RL 173).

Line 168-170 – I am lost here at what sea level change has to do with the climate- perhaps more explicitly link with winter temperatures and summer precipitation
Thank you. We revised this paragraph in response to this suggestion (RL 174-181)

Line 170 – ‘declines in delta18O correspond with rapid sea-level rise associated with meltwater

pulse 1b... which led to increased freshwater export from the Arctic Ocean into the North Atlantic known to influence AMOC stability’ – here again you should be more explicit about what you mean. Did the changes in SL and AMOC cause the delta18O decline? And if so how? This was a comment raised in my previous review which has not been improved by the re-write.
Thank you. We revised this paragraph in response to this suggestion to explicitly explain the relationship between $\delta^{18}\text{O}$ declines and sea level (RL 182-186)

Line 181 – I would combine this paragraph with the one above, where no results are mentioned currently
Thank you. We have significantly revised these paragraphs (RL187-208).

Line 217- ‘are understood to reflect winter precipitation due to groundwater processes’ – understood by who? A reference may be needed here. It is not clear why this interpretation is discounted
Thank you, we significantly revised this paragraph to clarify our interpretation of the Late Holocene data (RL 240-250).

Line 219 – ‘which also result in significant isotope depletion’ – I think references are needed here
References added (RL 244)

Line 220 – ‘circulation patterns are well recognized in driving basin-wide patterns in land and ocean temperatures’ – if this is the case, why do the late Holocene changes differ from each other?
Thank you for pointing out the need for better explaining this concept. We have revised this paragraph to explain that open and closed lake records that reflect sensitivity to winter and summer conditions, respectively, may differ in the same regions (see also our response to Major point #3, above). (RL 246-250)

Line 241 – ‘weak evidence’ – why is this data weak? if it is not reliable then it should be not used to argue for warm summers, as you do here.
Thank you, we revised this phrase to "the absence of glacial moraine evidence for advances during the YD chron" (RL266)

Line 387-389, “a fortuitous aquatic macrofossil...” I would suggest remove the “fortuitous” in case some readers interpret it as “lucky” or “fortunate”.
We deleted ‘fortuitous’ and this discussion was moved to the Supplementary section where we discuss the reservoir corrections for aquatic macrofossils.

The reference list is not fully formatted according to submission guidance. Please ensure consistency in reference style (e.g., author names, journal abbreviations, use of italics) and verify that all references are complete and correctly formatted. In addition, there are typographical errors: extra quotation marks at Lines 517, 576, and 583, and a line break issue at Line 533.

Thank you, we carefully revised the reference list following these observations.

Reviewer #3

We thank Reviewer #3 for their co-review.

Reviewer #4

The authors have provided a revised version of their manuscript which shows that similar decreases in $d18O$ values in two contrasting time intervals (the Younger Dryas and the Late Holocene) can be explained by different climatic mechanisms. In the former case, it is synchronization of Alaska with the North Atlantic Ocean, and in the latter attribution to changes in Pacific Ocean basin atmospheric circulation.

The conclusions remain plausible, and are important results.

I do feel that the manuscript makes a solid contribution to the potential causes of Holocene precipitation $d18O$ variations, but I don't feel that the manuscript conclusively demonstrated the attribution of the Late Holocene variations, and I was left feeling a little underwhelmed with the final conclusions.

The records themselves are really high quality and the paper could even emphasize more strongly (as the authors did in the response to reviewers) that these new records are among the most securely-dated in the circum-Arctic.

We are sincerely grateful to Reviewer #4 for their recognition of our findings, their quality, and their importance. We appreciate their excellent observations and have revised the data presentation to more conclusively demonstrate late Holocene attributions and more strongly emphasize the strengths of our data, including the final conclusions, described in detail below. We thank the reviewer for helping to guide these important improvements.

Minor comments:

There are a lot of typos in the manuscript. I didn't try to correct them all.

Thank you, we apologize. We have conscientiously reviewed the revised manuscript for typos.

The stable isotope terminology in several cases is incorrect, i.e., it is not possible to have "depleted isotopes". See Table 2.1 of Sharpe's Stable Isotope Geochemistry book for recommendations on proper terminology. "Absolute depletion of $d18O_p$ variations" also doesn't make sense. A sample can be depleted in $18O$, or depleted in $16O$, but it can't be deplete generally. There are other instances where improvement would help the manuscripts presentation.

Thank you, we agree, and have corrected every instance of stable isotope terminology according to the correct conventions.

Figure S2 is not legible, and several of the other Supp figures are of low to poor quality.

Thank you for identifying this problem. We have re-inserted all figures as .PNG images at 600dpi and hopefully this improves the resolution.

I find the authors explanation that meridional flow forced by atmospheric circulation changes in the Late Holocene could produce the very negative d18O anomalies. But I'm still not sure about the attribution. What forced the change? The authors suggested PDO/ENSO type variations, which are plausible sources of the anomalies, but I didn't see a strong conclusion, i.e., a link to a proxy record of PDO or ENSO that can convincingly show a correlation.

We deeply thank the reviewer for this suggestion, and we have accordingly revised Fig. 2 to show proxy reconstructions of ENSO and eastern equatorial Pacific SSTs. The combined ENSO records make a simpler and stronger visual comparison that convincingly shows a deglacial correlation between the North Pacific with Greenland and a late Holocene correlation with the Equatorial Pacific. We appreciate the reviewer for guiding this important improvement.

L35: I wouldn't state that lake carbonate and ice d18O are directly comparable, isotopically speaking. One is an indirect precipitation proxy, the other direct.

Thank you, 'directly' has been deleted (RL 43)

L39: delete "drift", an outdated term.

Thank you, this sentence has been deleted.

*L46-7, i think you mean Supp Fig 2? I have no problems with the authors *interpreting* the calcite d18O as a proxy for the d18O of the lake water, which reflects the d18O of winter precipitation. That is well supported. But whether the calcite is truly at equilibrium and has remained so "faithfully" is not so clear.*

Thank you – we have corrected this to Supplementary Fig 2. We have removed the word “faithfully” to acknowledge that equilibrium stability is difficult, or perhaps impossible, to prove. However, our multi-proxy data also shows no clear evidence for a large change in productivity across intervals of large $\delta^{18}\text{O}$ shifts, which the literature suggests are possible drivers of equilibrium instability (RL 64-66)

L48: How large is the region?

Added area of ~1500km² (RL 69)

L241: What does "weak evidence in glacial moraines" mean?

Thank you, we revised to ‘the absence of glacial moraine evidence for YD advances’ (RL 266)

L246: do you mean "Attribution of such shifts"?

Thank you, we revised this sentence for a different emphasis (RL 271)

L251: This sentence should be rewritten to something like "Our data show that similar decreases in lake sediment d18O during the YD and Late Holocene can be forced by different mechanisms." As written it doesn't make any sense, because fractionation is not what is being investigated here.

Thank you for this helpful clarification. We made this change in the conclusion (RL 277-278) and revised the abstract (RL 26-28). We thank the reviewer for guiding important improvements.

Response to Referee Comments (in blue text)

Reviewer #4 (Remarks to the Author):

The authors have done a nice job of improving the paper!

Thank you!

I am satisfied with the revisions and feel the manuscript is ready for publication, after one small change:

The references to $d_{18}O_p$ when referring to the lake sediment records is incorrect. They should be $d_{18}O_c$, which certainly tracks past changes in $d_{18}O_p$, but with a time-varying temperature-dependent fractionation offset and whatever lake-specific isotopic effects are happening. It is ok to use $d_{18}O_p$ when talking about inferences about changes in the $d_{18}O$ of past precipitation, but not when referring to the sediments.

We take your point and have revised the $d_{18}O$ subscripts of 'c' and 'p' in the main text following this suggestion. Changes were made on Lines 64, 66, 90, 93, 108, 113, and 178.